

**Residential burning is a significant source of soluble iron to the ocean**
Rui Li,[1,#,a] Haley E. Plaas,[2,#,b] Yifan Zhang,[1,3] Yizhu Chen,[1,3] Tianyu Zhang,[1,3] Yi Yang,[4] Sagar
Rathod,[5] Guohua Zhang,[1] Xinming Wang,[1] Douglas S. Hamilton,[2,*] Mingjin Tang[1,6,*]
[1] State Key Laboratory Advanced Environmental Technology and Guangdong Key Laboratory

7        of Environmental Protection and Resources Utilization, Guangzhou Institute of

8        Geochemistry, Chinese Academy of Sciences, Guangzhou, China

[2] Marine, Earth, and Atmospheric Sciences, North Carolina State University, Raleigh, NC,

10       USA

[3] College of Earth and Planetary Sciences, University of Chinese Academy of Sciences, Beijing,

12       China

[4] Key Laboratory of Geographic Information Science of the Ministry of Education, School of

14       Geographic Sciences, East China Normal University, Shanghai, China

[5] Department of Atmospheric, Oceanic, and Space Sciences, University of Wisconsin-Madison,

16       Madison, WI, USA

[6] Institute of Surface-Earth System Science, School of Earth System Science, Tianjin

18       University, Tianjin, China

[a] Current affiliation: School of Public Health, MOE Key Laboratory of Coal Environmental

21       Pathogenicity and Prevention, Shanxi Medical University, Taiyuan, China



[b] Current affiliation: Columbia University, Center for Climate Systems Research, New York,

23        NY 10025, USA; NASA Goddard Institute for Space Studies, New York, NY, USA

**Correspondence:**

Mingjin Tang (mingjintang@126.com)

Douglas S. Hamilton (dshamil3@ncsu.edu)

# The two authors contributed equivalently to this work.



**Abstract**

Understanding the physicochemical processes that supply atmospheric aerosol iron (Fe) to the ocean is crucial in our understanding of global biogeochemical cycles. Anthropogenic emissions contribute significant fluxes of aerosol Fe to the atmosphere, the soluble fraction of which can modulate marine primary productivity upon its deposition to the ocean surface. However, aerosol Fe solubility remains poorly constrained, due in part to a lack of direct measurements spanning a multitude of anthropogenic sources. We measured solubility of aerosol Fe from several distinct anthropogenic combustion processes and fuel types. The median Fe solubility varied widely by source, ranging from 0.03% for power plant coal fly ash to 55.87% for biofuel burning; furthermore, residential coal burning aerosol possessed much higher Fe solubility than industrial coal fly ash. Using new Fe solubilities reported herein, we updated parameters for anthropogenic aerosol Fe within the Mechanism of Intermediate complexity for Modeling Iron, an aerosol Fe module of the Community Earth System Model v2. Such updates led to significant improvement in model performance over ocean regions heavily influenced by anthropogenic emissions, and we identified residential burning as a significant source of soluble aerosol Fe to the ocean. Our work underscores the need to further refine understanding of physicochemical properties of aerosol Fe from a wide variety of anthropogenic sources. In turn, this understanding will aid in characterizing the influences of anthropogenic activities on past, present, and future atmospheric nutrient inputs to marine ecosystems.





## 1 Introduction

Anthropogenic activities have altered the atmospheric burden and deposition fluxes of biogeochemically relevant trace metals, including iron (Fe) (Bergas-Massó et al., 2023; Hamilton et al., 2020a). Fe availability in ocean waters plays a particularly important role in modulating the spatiotemporal distribution of primary productivity in ocean ecosystems, which has downstream impacts on marine fisheries and carbon sequestration (Ito et al., 2021; Tagliabue et al., 2014; Tagliabue et al., 2017). Energy-production, transportation, shipping, and manufacturing (e.g., steel production) are all characterized sources of anthropogenic aerosol Fe (Ito and Miyakawa, 2023; Ito and Shi, 2016; Rathod et al., 2024). These differing combustion fuel types possess distinct physicochemical properties that influence their impact on radiative forcing and nutrient supply (Al-Abadleh et al., 2023; Ito et al., 2018; Matsui et al., 2018; Rathod et al., 2020).

To assess the potential nutritional impact of atmospheric Fe deposition on ocean ecosystems, atmospheric aerosol research primarily focuses on tracing the soluble Fe content in aerosol (Baker et al., 2020; Ito et al., 2019; Mahowald et al., 2018). Soluble Fe content is often expressed as the fraction of soluble to total Fe in aerosol and then reported as a percentage solubility (Baldo et al., 2022; Liu et al., 2022; Mahowald et al., 2009). Several key processes control solubility of aerosol Fe over the course of its lifetime: 1) Fe mineralogy, 2) interactions with acidic and organic species in aerosol and cloud water, and 3) particle size and surface area to volume ratios (Bergas-Massó et al., 2023; Journet et al., 2008; McDaniel et al., 2019). Anthropogenic combustion not only alters the magnitude and spatial distribution of Fe fluxes to and from the atmosphere and surface ocean, but also influences the composition of the



atmosphere, that in turn, influences dissolution chemistry of aerosol Fe both directly and
indirectly. Mixing of aerosol Fe with acidic (e.g., sulfates or nitrates) and organic species (e.g.,
oxalate) co-emitted during anthropogenic combustion increases its solubility during transport
(Bergas-Massó et al., 2023; Chen et al., 2024; Itahashi et al., 2022; Li et al., 2017; Longo et al.,
2016). Furthermore, diverse technologies utilized during combustion processes (i.e., variable
combustion temperatures, boilers vs. furnaces, degree of emission control, and the fuel quality)
also influence the physicochemical properties of aerosol Fe beyond the composition of fuel
alone. As a result, study of how socioeconomic, technology, and policy driven changes to
anthropogenic fuel-burning, is needed to anticipate impacts on the global Fe cycle (Hamilton
et al., 2020a).

When compared to mineral dust, anthropogenic emissions of aerosol Fe are several orders

of magnitude lower at the global scale; however, anthropogenic Fe has a higher fractional
solubility (Ito et al., 2021) and source regions of dust and anthropogenic Fe are usually spatially
distinct (Hamilton et al., 2020a; Hamilton et al., 2019). Therefore, anthropogenic activity can
be a major contributor to Fe fluxes in many high nutrient low chlorophyll (HNLC) ocean
regions (Hawco et al., 2025; Liu et al., 2022).

Despite the importance of understanding anthropogenic Fe fluxes, the fractional solubility

of aerosol Fe, emitted from various anthropogenic sources, remains poorly understood
(Desboeufs et al., 2005; Li et al., 2022b; Oakes et al., 2012); consequently, Fe solubility
parameterizations in modeling studies for anthropogenic Fe vary widely (Ito et al., 2019;
Myriokefalitakis et al., 2018). Accordingly, in this work, we measured the Fe content and
solubility of aerosol emitted by several important anthropogenic sources (i.e. coal power plants,



steelwork industry, municipal water combustion, oil combustion, residential coal, and biofuel
burning). Then, using an Earth System Model, we applied novel observational findings by
updating Fe solubility parameters in distinct anthropogenic combustion fuel-sources. Modeled
Fe solubilities were validated against a global observational dataset at the regional scale. We
further used this model to quantify and bound uncertainties in emission and deposition fluxes
of soluble Fe under three anthropogenic combustion emission scenarios spanning past (pre-
industrial) to future (shared socioeconomic pathway 3-7.0) conditions.

## 2 Methodology

Experimental and modelling methods employed in our work are described in Sections 2.1
and 2.2, respectively.

### 2.1 Experimental methods

This work examined six types of anthropogenic combustion aerosol, which were classified
into two broad categories. The first category, fly ash, included power plant coal fly ash,
steelwork fly ash, municipal waste fly ash, and oil fly ash. The second category, residential fuel
sources, included residential coal and biofuel combustion. Biofuels examined in this work were
limited to straw, wood, grasses and leaves, and we did not examined other biofuels such as
dung. We also studied one oil bottom ash sample, in order examine whether Fe solubility is
significantly different between oil fly ash and bottom ash.

### 2.1.1 Fly ash and bottom ash samples

Power plant coal fly ash samples were obtained from electrostatic precipitators or
baghouse rows in coal power plants in 29 provinces in China (Li et al., 2021; Liu et al., 2021);
one coal power plant was selected in each province except for Guangdong and Shandong where



two coal power plants were selected for each province. As a result, 31 power plant coal fly ash
samples were examined in total. In addition, we examined 29 steelwork fly ash samples
collected from different iron and steel plants, three municipal waste fly ash samples, two oil
fly ash samples (Wu et al., 2018) and one oil bottom ash sample (Fu et al., 2012). Fly ash and
bottom ash samples were directly provided by these factories.
Fly ash and bottom ash samples (~10 mg for each sample) were digested and then
analyzed using inductively coupled plasma mass spectrometry (ICP-MS) to determine their Fe
content. Experimental procedures for sample digestion and total Fe measurement can be found
elsewhere (Li et al., 2022c). Soluble Fe was leached and determined using the procedure
described in our previous work (Li et al., 2022b). In brief, fly ash and bottom ash samples (~20
mg for each sample) were individually leached in 20 mL sodium acetate buffer (5 mmol/L, pH
= 4.3) for 2 h, during which an orbital shaker (300 rpm) was used to stir the solution. The
aqueous mixture was centrifuged (3000 rpm) for 15 min, and a pH paper (range: 3.5-6.8;
precision: 0.3; Macherey-Nagel, Germany) was used to measure the pH of the solution and no
measurable pH change occurred after leaching. The aqueous solution was filtered through a
polyethersulfone filter (pore size: 0.22 μm), acidified to contain 1% (v/v) nitric acid and then
analyzed by ICP-MS to measure soluble Fe. In this work, fractional solubility of Fe was
reported as the ratio (in %) of soluble Fe to total Fe.
**2.1.2 Residential coal and biofuel combustion aerosols**
Generation and collection of residential coal and biofuel combustion aerosols are detailed
in the supplement (Text S1). In brief, we burned coal and biofuel in a commercial cook stove
which is widely used in rural areas in China, and collected $PM_{2.5}$ samples (aerosol particles

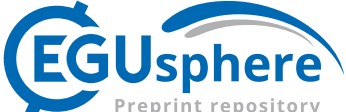

with aerodynamic particle diameters below 2.5 μm) onto pre-cleaned Whatman 41 (W41)
cellulose filters using a medium volume aerosol sampler (TH-150C, Tianhong Co.).

Our work examined three types of coal (anthracite, semibituminous coal, and bituminous

coal) and nine types of biofuel (wheat straw, rice straw, corn straw, rape straw, cogongrass,
China fir trunk, pine trunk, poplar trunk, and pine needle) commonly found in China. We
collected eight filter samples for each fuel type, except anthracite for which we only collected
two filter samples. We had to combine some filter samples in our experimental analysis to meet
the detection limit for soluble Fe; as a result, the number of effective filter samples (for which
Fe content and solubility were reported) were usually <8 for each fuel type.

After aerosol collection, the filters were individually placed in a pre-cleaned Petri dish

and then stored in a desiccator for 60 h to remove particle-associated water. The mass of filters
before and after aerosol collection were measured (accuracy of 0.1 mg), and the mass of
particles collected ranged from 2.5 to 432.7 mg. Each filter was then divided into two equal
parts. To determine the soluble Fe content, the first half of a filter was leached in 20 mL sodium
acetate buffer (5 mmol/L, pH = 4.3) for 2 h (Section 2.1.1) and analyzed using ICP-MS. Fe
concentrations in some leaching solutions were low; as a result, these leaching solutions (~15
mL for each solution) were combined for the same fuel type and then pre-concentrated to a
volume of 6 mL, in order to increase Fe concentration in the solution used for ICP-MS analysis.
The second half of a filter was digested and analyzed by ICP-MS to determine total Fe, and the
experimental procedure used can be found in our previous work (Zhang et al., 2022). If leaching
solutions were combined for the first parts of these filters, their second parts were also
combined and digested together to allow direct comparison.



**2.2 Model simulations**
**2.2.1. Atmospheric Fe model description**
Earth System Models can investigate the spatiotemporal distribution and fluxes of key
atmospheric nutrients under various climatological regimes (Hamilton et al., 2020a; Hamilton
et al., 2022; Wu et al., 2020). To test the impact of new soluble Fe parameters on modeled
fluxes of soluble aerosol Fe to the atmosphere and marine ecosystems, we used the Mechanism
of Intermediate complexity for Modeling Iron (MIMI). MIMI is an Fe aerosol-chemistry
module embedded within the atmospheric component (Community Atmosphere Model version
6, CAM6) of the Community Earth System Model version 2 (CESM2) (Danabasoglu et al.,
2020; Hamilton et al., 2019). Mineral dust, anthropogenic combustion, and wildfire emissions
are currently represented as sources of aerosol Fe in MIMI. The current dust emission scheme
within MIMI includes an updated soil moisture submodule within the land component of the
model that prognostically calculates dust aerosolization as a function of soil moisture (Li et al.,
2022a). The inclusion of these improvements to dust and updated anthropogenic Fe sources
represents a new working version of MIMI v1.1, as described herein.
A comprehensive overview of MIMI model details and parameters is provided in
Hamilton et al. (2019), and in brief, MIMI simulates the emission, atmospheric transport, and
deposition of Fe-containing aerosol within three distinct particle size modes (Aitken,
accumulation, and coarse modes). Within each source of aerosol Fe (dust, wildfire, and
anthropogenic combustion), both the insoluble and soluble fractions are carried as separate
tracers, and the soluble fraction of Fe for each aerosol source is assigned at the point of emission.
Prior to deposition and during atmospheric transport, Fe solubility is further modified via non-



reversible multiphase reactions with acidic and organic species. Acidic processing is a function
of aerosol pH and temperature, while organic processing is an aqueous phase chemistry
reaction that depends on oxalate concentrations which are calculated based on the
concentrations of secondary organic aerosol present (Johnson and Meskhidze, 2013; Scanza et
al., 2018).

The model is gridded in a 3-dimensional space at a resolution of 0.96×1.25 degrees

(latitude × longitude) and includes 56 vertical pressure levels from the surface to 2 hPa at the
highest altitude. Meteorology is forced in all the simulations using Modern-Era Retrospective
analysis for Research and Applications Version 2 (MERRA-2), and a 1-year model spin up
was undertaken for all simulations.

**2.2.2 Global pyrogenic Fe emission inventories and input dataset development**

While dust Fe emissions are calculated prognostically within MIMI, anthropogenic and

wildfire (pyrogenic) emissions were prescribed using emissions inventories. To supply initial
anthropogenic Fe emission fluxes to the model, we used an annual mean emission inventory
for anthropogenic combustion Fe that was first developed in Rathod et al. (2020) and further
detailed in Rathod et al. (2024). In this inventory, Fe content in combustion aerosol was
empirically derived for the present day (PD; climatological year 2010) using the Speciated
Pollution Emissions Wizard (SPEW) (Bond et al., 2007; Bond et al., 2004) which characterized
anthropogenic emissions of particulate matter by fuel-source and combustion technology.
Soluble and insoluble Fe content in emissions were dependent on fuel-type and were
accordingly segregated by key sectors and fuels: 1) industrial fossil fuel (coal), 2) industrial
and vehicular fossil fuels (oil), 3) smelting operations (steel/iron), and 4) residential





cooking/heating (biofuel/biomass/wood) (Rathod et al., 2020). Industrial oil emissions were
separated by land- and sea-based emissions to distinguish terrestrial transportation from
shipping. Wildfire-Fe emission parameters are detailed in Hamilton et al. (2019), and in this
work we use the CMIP6 (Coupled Model Intercomparison Project phase six) fire emission
datasets for PD simulations (van Marle et al., 2017).

For the first time, we segregated anthropogenic coal Fe sources into industrial and

residential coal burning sources. In the dataset provided by Rathod et al. (2020), Fe from the
coal source was assumed to only stem from industrial activity, but here we isolate the
residential source. This segregation was achieved by applying a spatially resolved (on the Earth
System Model grid box co-ordinates) ratio of industrial-to-residential coal that was calculated
based on the ratio of industrial-to-residential black carbon (BC) emissions within the CMIP6
dataset (Hoesly et al., 2018), and this calculation assumed that the Fe-to-BC ratios were
matched between sources.
**2.2.3 Model simulations performed**

Ten model simulations were performed to evaluate the impact of changes to Fe solubility

of anthropogenic combustion aerosol (Section 2.1) on atmospheric soluble Fe fluxes to key
marine ecosystems (Table 1). For all simulations, we set the model climatology to present-day
(PD) conditions spanning 2009-2011. Simulations were distinguished as cases (with variable
Fe solubility parameterizations) within different emission scenarios (with variable
anthropogenic combustion emission fluxes).



**Table 1.** Description of model simulations performed using MIMI with emission scenarios and
emissions inventory (database). PD = present day (2010 CE), PI = pre-industrial (1750 CE),
SSP370 = shared socioeconomic pathway scenario 3-7.0, MID = midcentury (2040-2050 CE)
and END = end century (2090-2100 CE). NA = assumed industrial activity is zero at 1750 CE.

| Emissions Scenario | Simulation | BC Emissions database | BC Emission (Tg a$^{-1}$) |
|---|---|---|---|
| PD | PD-BASE | CMIP6 | 6.46 |
| PD | PD-RESI | CMIP6 | 6.46 |
| PD | PD-BIOF | CMIP6 | 6.46 |
| PD | PD-IND | CMIP6 | 6.46 |
| PI | PI-BASE | NA | NA |
| PI | PI-RESI | NA | NA |
| FU (2050) | MID-SSP370-BASE | SSP3.70 | 8.30 |
| FU (2050) | MID-SSP370-BIOF | SSP3.70 | 8.30 |
| FU (2100) | END-SSP370-BASE | SSP3.70 | 6.33 |
| FU (2100) | END-SSP370-BIOF | SSP3.70 | 6.33 |


Four PD simulations aimed to assess the impact of each new solubility parameter on the

ability of the model to capture ship-based observations of total Fe, soluble Fe and Fe solubility.
These simulations utilized the anthropogenic combustion emission inventory representing PD
emissions as described in Section 2.2.2. The first PD case (PD-BASE) served as a baseline, i.e.
no changes were made to anthropogenic Fe solubilities when compared to previous studies
using MIMI to model global Fe fluxes (Rathod et al., 2020; Rathod et al., 2024). In the next
three PD cases (PD-RESI, PD-BIOF, PD-IND), fractional solubility was updated incrementally
for individual sectors to assess fuel-type specific impacts to soluble Fe fluxes, which are later
detailed in Section 3.3. Information on model validation and constraint to ship-based
observations of aerosol Fe is provided in Section 2.2.5.





Using both pre-industrial (PI; 1750 CE) and future (FU; 2050 and 2100 CE) anthropogenic
emissions scenarios, we performed six additional model simulations using the -BASE and -
BIOF solubility parameters applied during the PD simulations; these solubility cases were
selected based on model validation results as presented in Section 3.3. To isolate how changes
in soluble aerosol Fe fluxes responded to changes in emission parameterizations and
subsequent dissolution chemistry, PI and FU simulations were conducted with meteorological
and climatological conditions identical to PD (2009-2011). Emissions inventories (including
Fe) were modified to represent PI and future FU projections for atmospheric emissions.
PI-BASE and PI-BIOF served as a baseline for comparison to PD and FU simulations,
with minimal influence on the Fe cycle by anthropogenic emissions (Table 1). In the PI era, Fe
emissions were predominantly natural in source (e.g., dust and wildfire), and only residential
biofuel burning was represented as an anthropogenic source of Fe (Hamilton et al., 2020b). All
other anthropogenic sources of Fe were assumed to be zero.
MID-SSP370-BASE, MID-SSP370-BIOF, END-SSP370-BASE, and END-SSP370-
BIOF were conducted to evaluate the projected impact of socioeconomic changes to energy
production and fuel-usage in communities across the globe, as well as population increases.
We chose to utilize the emissions scenario detailed in the Shared Socio-economic Pathway
3-7.0 "regional rivalry" (SSP370), which represents anticipated sociopolitical and
environmental changes resulting in an increase to radiative forcing by 3-7.0 W m$^{-2}$ by the year
of 2100 (Riahi et al., 2017). In this scenario, air pollutants sourced from anthropogenic activity
are projected to be highest when compared to alternative SSP scenarios (SSP126, SSP245, and
SSP585). Accordingly, model simulations using SSP370 can be leveraged to establish an upper



bound estimate for future soluble aerosol Fe fluxes from anthropogenic combustion emissions.
Dust fluxes in future Fe emission scenarios were adjusted to account for dust-climate feedback
using a scaling factor ranging between 1.0-1.1, as described in Hamilton et al. (2020b). Given
that BC emissions are anticipated to peak in the midcentury (2040-2050) but return to PD-
comparable emissions by 2100 (Turnock et al., 2020), we assessed projected changes to Fe
emissions at both the mid-point (2050) and endpoint of the 21$^{st}$ century (2100).
**2.2.4 Preindustrial (PI) and future (FU) Fe emission estimates**
For PI simulations, we used a pre-developed Fe combustion emission inventory (Hamilton
et al., 2020a). Only residential biofuel burning served as an anthropogenic source of Fe due to
a presumable lack of industrialized anthropogenic emissions (i.e., fossil fuels and smelting);
global emission was $0.7 \times 10^{-3}$ Gg Fe a$^{-1}$ and only occupied the fine aerosol mode (i.e., sum of
Aitken and accumulation modes). Details on the development of the PI Fe combustion emission
inventory are provided in Hamilton et al. (2020a). The SimFire inventory, coupled to the LPJ-
GUESS (Lund-Potsdam-Jena General Ecosystem Simulator) vegetation model, was used to
prescribe wildfire-Fe emissions during the PI era (Hamilton et al., 2018; Hamilton et al., 2020a;
Knorr et al., 2016).
For FU simulations, we developed two new Fe emissions datasets which were derived
from our PD dataset and linearly scaled for all combustion sources according to projected
changes in anthropogenic BC emissions via CMIP6 anthropogenic BC emission dataset
(Hoesly et al., 2018; Riahi et al., 2017). In the CMIP6 BC inventory, emissions are segregated
by the following sectors: agriculture, energy, industrial, terrestrial transportation,
residential/commercial/other, solvents production and application, waste, and international



288 shipping. We treated 'energy' and 'industrial' sources together as industrial coal BC sources

289 and ocean-masked 'international shipping' with land-masked 'terrestrial transportation'

290 together as oil BC sources. BC emissions labeled 'residential, commercial and other' were

291 separated into residential coal and residential biofuel sources of BC based on the grid-cell

292 specific ratios of residential coal Fe to residential biofuel Fe in our PD Fe emissions dataset.

293 BC emissions from smelting operations were not directly available for PI or FU projections;

294 therefore, they were set to 0 in the PI and maintained at PD levels in the FU.

295  Once FU BC emissions were organized according to combustion fuel-sources as

296 characterized herein, using a dynamic ratio of Fe-to-BC dependent on region, fuel-source

297 emission sector, and aerosol size fraction, we calculated scenario-specific Fe emissions in

298 individual grid cells using Eq. (1):

$$\frac{[Fe_X]_{i,j,a,b}}{[BC_X]_{i,j,a,b}} = \frac{[Fe_{PD}]_{i,j,a,b}}{[BC_{PD}]_{i,j,a,b}} \qquad (1)$$

300 where X denotes the emissions scenario (MID-SSP370 or END-SSP370), $i$ and $j$ represent the

301 longitudinal and latitudinal coordinates, $a$ represents the aerosol mode (fine or coarse), $b$

302 represents the fuel-source (industrial oil, industrial coal, residential coal, residential biofuel, or

303 smelting), and [Fe] and [BC] represent the fluxes (kg m$^{-2}$ s$^{-1}$) of Fe and BC, respectively. Using

304 these gridded emissions, fuel-sources were then summed and segregated by soluble fraction to

305 generate six combustion-Fe tracers to be read into and transported within the model,

306 distinguished by aerosol size and solubility as calculated using Eqs. (2-3):

$$[Fe_{insol}]_a = \Sigma \{[Fe_X]_{i,j,a,b} * (1 - sol_b)\} \qquad (2)$$

$$[Fe_{sol}]_a = \Sigma ([Fe_X]_{i,j,a,b} * sol_b) \qquad (3)$$



where *insol* represents the insoluble fraction, *sol* represents the soluble fraction, and $sol_b$
represents the fractional solubility for each fuel-source (*b*). As a final step, the fine mode was
split into accumulation and Aitken modes by applying a simple ratio of 9:1.

For wildfire-Fe emissions in FU scenarios, we used the CMIP6 fire emission datasets for

each respective simulation, i.e. MID-SSP370 and END-SSP370 (Bergas-Masso et al., 2025;
Hamilton et al., 2024).
**2.2.5 Model validation**

To evaluate model performance, we compared global observations of total Fe

concentration, soluble Fe concentration, and Fe solubility to modeled values for each PD
simulation, grouping data by key aerosol deposition and ocean biogeochemistry regions. The
observational dataset of Fe content in aerosol was reported in Hamilton et al. (2019) and
updated herein to include measurements published between 2021 and 2024 (n = 1624).
Observed Fe solubility in aerosol spans five orders of magnitude (Perron et al., 2024), and one
reason for this large range is due to differences in experimental procedures during
quantification (Tang et al., 2025). To facilitate a more direct comparison between modelled
and observed soluble Fe content, we removed observations from the global dataset that did not
measure soluble Fe directly. When multiple observations fell within a model grid cell, values
were aggregated to climatological averages, using medians to be most representative of
expected variations in Fe fluxes across time and space (final n = 990; Figure S3). For final
evaluation of the model capability in simulating surface Fe concentrations, both model and
observational data were grouped into key ocean regions (Figure 1), based on predominant



sources of atmospheric aerosol and phytoplankton nutrient limitation dynamics (i.e., HNLC
regions) as revealed in Hamilton et al. (2019) and Hamilton et al. (2023).

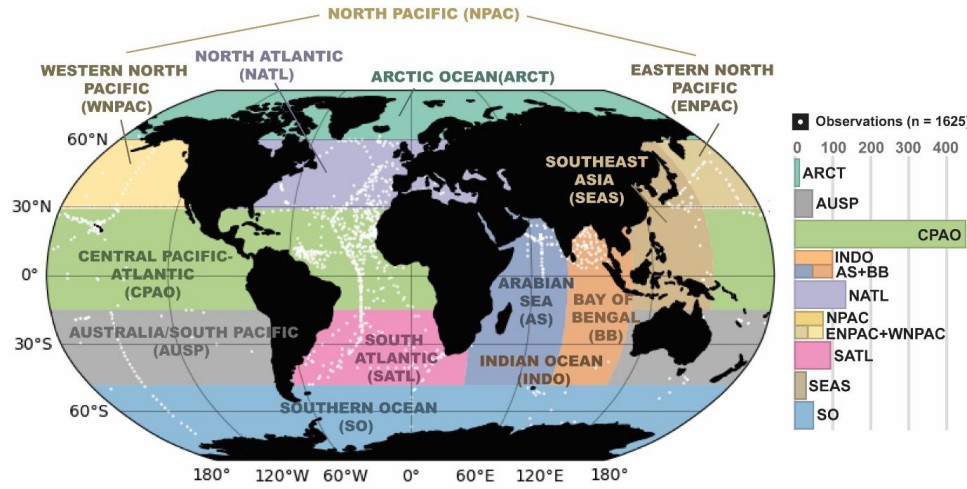


**Figure 1.** Regional groupings for model-observation comparisons of surface Fe concentrations
(ship-based, in aerosol). The coordinates for individual Fe observations are indicated with a
white circle. Number of soluble Fe observations within each region are provided by the
histogram.

**3 Results and Discussion**
Sections 3.1 and 3.2 present Fe content and solubility measured in our experimental work,
and modeling results are presented in Section 3.3.
**3.1 Fe content by fuel type**
This work quantified the Fe content in particles from six different combustion and
anthropogenic sources, including power plant coal fly ash, residential coal combustion aerosol,
steelwork fly ash, residential biofuel burning aerosol, municipal waste fly ash, and oil fly ash
(Table 2; Fe content in individual samples is provided in Tables S1-S5).





**Table 2.** Summary of Fe content and solubility for power plant coal fly ash, residential coal
combustion aerosol, steelwork fly ash, biofuel burning aerosol, municipal waste fly ash, oil fly
ash and oil bottom ash examined in our work (*n*: number of samples examined in our work).

| sample type | *n* | range | average | median |
|---|---|---|---|---|
| Fe content (mg/g) | | | | |
| power plant coal fly ash | 31 | 20.7-103.8 | 37.2±16.8 | 35.0 |
| residential coal combustion aerosol | 10 | 0.025-0.101 | 0.044±0.023 | 0.038 |
| steelwork fly ash | 29 | 5.8-918.9 | 312.6±246.1 | 346.5 |
| biofuel burning aerosol | 27 | 0.002-0.101 | 0.023±0.026 | 0.013 |
| municipal waste fly ash | 3 | 3.9-29.7 | 18.7±13.3 | 22.6 |
| oil fly ash | 2 | 9.1-18.3 | 13.7±4.6 | 13.7 |
| oil bottom ash | 1 | - | 191 | 191 |
| Fe solubility (%) | | | | |
| power plant coal fly ash | 31 | 0.002-0.17 | 0.05±0.05 | 0.03 |
| residential coal combustion aerosol | 10 | 7.03-100 | 33.30±27.71 | 28.45 |
| steelwork fly ash | 29 | 0.007-10.64 | 1.37±2.77 | 0.07 |
| biofuel burning aerosol | 28 | 2.86-100 | 56.07±30.95 | 55.87 |
| municipal waste fly ash | 3 | 0.58-2.41 | 1.51±0.92 | 1.54 |
| oil fly ash | 2 | 11.70-13.43 | 12.56±0.87 | 12.56 |
| oil bottom ash | 1 | - | 25.47 | 25.47 |





### 3.1.1 Power plant coal fly ash

Fe content ranged from 20.7 to 103.8 mg/g for the 31 power plant coal fly ash samples examined in our work, with average and median values being $37.2 \pm 16.8$ and 35.0 mg/g, respectively. As shown in Table S6, Fe content ranged from 16.0 to 52.0 mg/g (n = 3) in one study (Baldo et al., 2022), with mean and median values being $33.0 \pm 18.0$ and 31.0 mg/g; in another study (Goodarzi, 2006), the median value of Fe content was determined to be 34.4 mg/g (n = 7). Fe content measured by these two studies (Baldo et al., 2022; Goodarzi, 2006) agreed well with our work. Some other studies (Dutta et al., 2009; Fu et al., 2012; Jankowski et al., 2006; Meij, 1994) found higher mean or median Fe content for power plant coal fly ash (Table S6), but the reported ranges overlapped with our work. For example, Fe content were found to range from 38.3 to 98.6 mg/g (n = 7) in one study (Li et al., 2022b), with mean and median values being $62.1 \pm 26.7$ and 43.2 mg/g; in another study (Moreno et al., 2005), Fe content were found to range from 18.2 to 112.0 mg/g (n = 23), with mean and median values being $57.8 \pm 22.7$ and 52.5 mg/g.

In summary, the mean or median Fe content reported in different studies are typically in the range of 30-70 mg/g for power plant coal fly ash, and this variability is likely due to difference in coal (Wang et al., 2015; Ward, 2016) and combustion conditions (Blissett and Rowson, 2012; Kutchko and Kim, 2006). Fe content in power plant coal fly ash was set to ~70 mg/g in some modeling studies (Luo et al., 2008; Rathod et al., 2020), being consistent with experimental results.



### 3.1.2 Residential coal combustion aerosol

For the 10 residential coal combustion aerosol samples ($PM_{2.5}$) we examined, Fe content ranged from 0.025 to 0.101 mg/g (Table 2), with average and median values being $0.044 \pm 0.023$ and 0.038 mg/g, respectively. Only a few previous studies measured Fe content in residential coal combustion aerosols (Table S6). The average Fe content was determined by Patil et al. (2013) to be $0.048 \pm 0.035$ mg/g (n = 3) for $PM_{2.5}$ and $0.061 \pm 0.044$ mg/g (n = 3) for $PM_{10}$, being similar to or slightly higher than our result. In another two studies (Watson et al., 2001; Zhang et al., 2012), the average Fe content was measured to be $0.671 \pm 0.023$ mg/g (n = 4) and $0.7 \pm 0.1$ mg/g (n = 5), significantly higher than our result, and such differences may be attributed to variations in coal types and combustion conditions. Overall, our and previous studies suggest that the Fe content in residential coal combustion aerosols is very low, typically below 1 mg/g. Fe content were set to 1 and 0.5 mg in previous modeling studies (Luo et al., 2008; Rathod et al., 2020), being broadly consistent with experimental results.

Fe content in power plant coal fly ash is much higher than residential coal combustion aerosols, primarily due to differences in combustion conditions (Rathod et al., 2020). Power plant coal fly ash has very low carbon content and is mainly composed of metals and minerals (Ahmaruzzaman, 2010; Li et al., 2022c; Patil et al., 2013); in contrast, residential coal combustion aerosol particles contain a large fraction of carbonaceous materials due to incomplete combustion, and thus the content of metals, including Fe, are much lower (Patil et al., 2013; Zhang et al., 2012). Furthermore, combustion temperature typically ranges from 1200 to 1700 ℃ for coal-fired power plant, enabling Fe in coal to enter fly ash particles through volatilization-condensation (Blissett and Rowson, 2012); residential coal combustion occurs at




much lower temperatures which are insufficient for Fe to enter aerosols through this process
(Rathod et al., 2020), also leading to lower Fe content.
**3.1.3 Steelwork fly ash**

For the 29 steelwork fly ash samples we examined, Fe content ranged from 5.8 to 918.9

mg/g, with mean and median values measured to be $312.6 \pm 246.1$ and 346.5 mg/g, respectively
(Table 2). As shown in Table S6, some previous studies have reported average Fe content to
be 358.9 (n = 1), 369.3 (n = 1), 312.2 (n = 1), and $329.1 \pm 22.6$ mg/g (n = 4) (Alizadeh and
Momeni, 2016; Silva et al., 2019; Souza et al., 2010; Vieira et al., 2013), in good agreement
with our results. Lower Fe content was also reported by previous work, with average values
being 86.0 (n = 1), 128.1 (n = 1), 150.8 (n = 1), 286.5 (n = 1), 284.6 (n = 1), 238.7 (n = 1), and
$267.3 \pm 4.8$ mg/g (n = 4) (Al-Negheimish et al., 2021; Alsheyab and Khedaywi, 2016; Laforest
and Duchesne, 2006; Li et al., 2023; Loaiza et al., 2017; Stathopoulos et al., 2013; Xia and
Picklesi, 2000); in contrast, some previous studies also found the average or mean Fe content
to be around 400-500 mg/g (Machado et al., 2006; Patil et al., 2013; Ye et al., 2021), slightly
higher than our results.

Despite some variability in Fe content reported by our and previous studies (Table S6),

the mean or median Fe content are generally around 300–500 mg/g for steelwork fly ash. In a
recent modeling study (Rathod et al., 2020), the Fe content in steelwork fly ash was set to 440
mg/g (and the lower and upper bounds were set to 150 and 950 mg/g), being consistent with
experimental results.



### 3.1.4 Biofuel burning aerosol

Our work considered biofuel burning aerosols for nine types of biofuels, including four types of crop straw, one type of wild grass, and four types of wood. Fe content in biofuel burning aerosols ranged from 0.002 to 0.101 mg/g (Table 2), with average and median values being $0.023 \pm 0.026$ and 0.013 mg/g, respectively. As shown in Table S6, the average Fe content was determined to be $0.024 \pm 0.017$ mg/g (n = 3) for $PM_{2.5}$ (Patil et al., 2013), very close to our result; in another study (Hildemann et al., 1991), it was determined to be 0.090 mg/g for $PM_2$ (n = 2), higher than our result. In some other studies, average Fe content were reported to be in the range of 0.162-0.440 mg/g for $PM_{2.5}$ (Alves et al., 2011; Hedberg et al., 2002; Watson et al., 2001; Zhang et al., 2012) and $0.723 \pm 0.661$ mg/g for $PM_{10}$ (Schmidl et al., 2008), much higher than our results.

Fe content in biofuel burning aerosols showed large variability in different studies, likely due to variations in combustion conditions and biofuel types. For example, metal content in biofuel burning aerosols depended greatly on biofuel types and regions where biofuel was collected (Goncalves et al., 2010), and aerosol particles emitted by wild grass combustion contained larger amounts of metal than wood combustion (Jahn et al., 2021). Modeling studies have used a similar distribution of Fe content between 0.2 and 0.580 mg/g for biofuel burning aerosols (Luo et al., 2008; Rathod et al., 2020).

### 3.1.5 Municipal waste fly ash, oil fly ash and oil bottom ash

For the three municipal waste fly ash samples we investigated, Fe content ranged from 3.9 to 29.7 mg/g, with average and median values being $18.7 \pm 13.3$ and 22.6 mg/g (Table 2). Several previous studies measured Fe content in municipal waste fly ash (Table S6). For

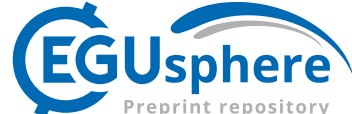

example, the average Fe content were measured to be 18.0 ± 13.3 mg/g (n = 3) and 23.1 mg/g
(n = 1) in two studies (Cobo et al., 2009; Raclavská et al., 2017), very similar to our results;
another four studies (Funari et al., 2017; Liu et al., 2009; Wu and Ting, 2006; Wu et al., 2012)
reported lower Fe content, ranging from 5.2 to 10.9 mg/g; some other studies (Bayuseno and
Schmahl, 2011; Lin et al., 2003; Wan et al., 2006; Zhang et al., 2011) also reported slightly
higher Fe content, ranging from 27.1 to 34.3 mg/g. In summary, most studies suggest that Fe
content in municipal waste fly ash are around 20 mg/g, and it has been set to 18.8 mg/g in a
modeling study (Rathod et al., 2020), being consistent with experimental results.
Fe content in the two oil fly ash samples we examined were measured to be 9.1 and 18.3
mg/g, and the average value was determined to be 13.7 ± 4.6 mg/g. The Fe content was
measured to be 15.0 mg/g for one oil fly ash sample (Celo et al., 2015), close to the value we
reported. In another two studies (Agrawal et al., 2008; Sippula et al., 2014), the average Fe
content was measured to be 1.98 ± 0.35 (n = 4) and 1.60 ± 1.21 mg/g (n = 14), lower than our
result. In a modeling study (Rathod et al., 2020), the Fe content was set to 10 mg/g for oil fly
ash, being consistent with the experimental results reported by our work and Celo et al. In
addition, in our work the Fe content was measured to be 191 mg/g for one heavy oil bottom
ash sample, much higher than that for oil fly ash.
**3.1.6 Fe contents: comparison of anthropogenic and dust Fe**
Figure 2a displays Fe content for anthropogenic particles examined in our current study,
and the brown dashed line represents the average Fe content of desert dust (35 mg/g) (Taylor
and McLennan, 1995). Steelwork fly ash has very high Fe content (median: 346.5 mg/g), about
one order of magnitude higher than desert dust. Power plant coal fly ash (median: 35.0 mg/g)



has similar Fe content to desert dust, while municipal waste fly ash (median: 22.6 mg/g) and
oil fly ash (median: 13.7 mg/g) have relatively lower Fe content than desert dust. Compared to
desert dust, Fe content were around three orders of magnitude lower for residential coal and
biofuel burning aerosol (median: 0.038 and 0.013 mg/g, respectively). The Fe content was
much lower for residential coal and biofuel burning aerosol, likely due to lower combustion
temperatures. When combustion occurs at lower temperature, the carbon content of emitted
particles is higher; in addition, lower combustion temperature is not sufficient to enable Fe in
the fuel to enter emitted particles via volatilization-condensation processes.

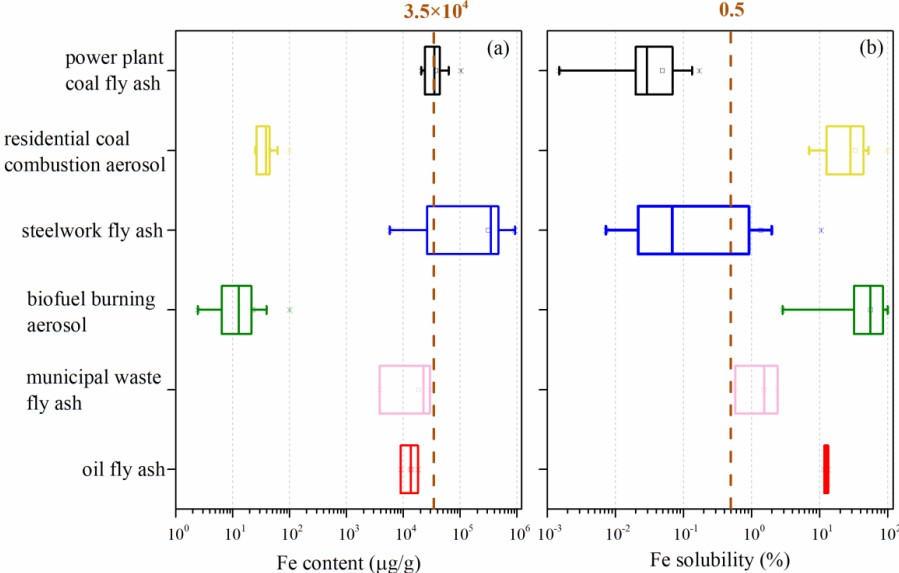


**Figure 2.** Fe content (a) and solubility (b) measured in our work for power plant coal fly ash,
residential coal combustion aerosol, steelwork fly ash, biofuel burning aerosol, municipal
waste fly ash and oil fly ash. The two brown dash lines represent (a) the Fe content ($3.5 \times 10^4$
μg/g) and (b) Fe solubility (~0.5%) for desert dust, respectively.



**3.2 Fe solubility by fuel type**

**3.2.1 Power plant coal fly ash**

Fe solubility in acetate buffer (pH: 4.3) was found to range from 0.002% to 0.17% for power plant coal fly ash (Table 2), with the average and median values being $0.05 \pm 0.05\%$ and 0.03%, respectively. A few previous studies measured Fe solubility of power plant coal fly ash in weakly acidic or circumneutral solutions (Table S7). For example, Fe solubility was measured to be 0.06% in deionized water (Oakes et al., 2012), similar to our result; it was measured to be 0.2% in dilute sulfuric acid solution (pH: 4.7) (Desboeufs et al., 2005), slightly higher than our result; the median Fe solubility was determined to be 0.13% in acetate buffer (pH: 4.3) and 0.06% in deionized water (Li et al., 2022b), both higher than the median value we obtained. Overall, our work and previous studies suggest that Fe solubility is low in weakly acidic and circumneutral solutions for power plant coal fly ash, with mean or median values around 0.1%.

Some studies also measured Fe solubility of power plant coal fly ash in highly acidic solutions and found them to be much higher than those in weakly acidic and circumneutral solutions. For example, Fe solubilities were found to be in the range of 20-25% at pH of 1-2 (Chen et al., 2012), 4.2-8.3% at pH of 2 (Fu et al., 2012), and >40% at pH of 2.1 (Baldo et al., 2022). Although Fe solubility measured in strongly acidic solutions may not reflect initial Fe solubility, these studies suggested that acid processing in the emission plume or wider atmosphere could greatly increase Fe solubility for power plant coal fly ash.



### 3.2.2 Residential coal combustion aerosol

Fe solubility in acetate buffer (pH: 4.3) was determined to range from 7.03% to 100% for residential coal combustion aerosol (Table 2), with the average and median values being 33.30 ± 27.71% and 28.45%, respectively. To our knowledge, no previous study has measured Fe solubility for residential coal combustion aerosol. Compared to power plant coal fly ash, Fe solubility was much higher for residential coal combustion aerosol, and such difference can be attributed to much higher temperature in power plant coal combustion than residential coal combustion. Pyrite ($FeS_2$) is the major Fe-containing mineral in coal (Deng et al., 2015; Oliveira et al., 2016; Rathod et al., 2020). In low-temperature combustion, pyrite is mainly transformed to Fe sulfate (Bhargava et al., 2009) which has very high Fe solubility; as the temperature increases to >1000 K, Fe sulfate is further transformed to hematite and magnetite which exhibit very low solubility (Hu et al., 2006; Ram et al., 1995; Rathod et al., 2020). A previous study (Rathod et al., 2020) used the relationship between combustion temperature and Fe mineralogy in emitted particles to estimate Fe solubility for different combustion aerosols, and Fe solubility was estimated to be as high as ~32.5% for residential coal combustion aerosols, in good agreement with our experimental results.

### 3.2.3 Steelwork fly ash

Fe solubility in acetate buffer (pH: 4.3) was determined to range from 0.01% to 10.64% for steelwork fly ash (Table 2), and the average and median values were 1.37 ± 2.77% and 0.07%, respectively. We note that Fe solubility was significantly higher (0.92%-8.59%) for 8 samples and very low (<0.5%) for the other 21 samples (Table S3), most of which showed Fe solubility below 0.1%. No previous work has measured Fe solubility for steelwork fly ash. Our



experimental results were supported by a modeling study (Rathod et al., 2020) which suggested

that the major Fe-containing species in steelwork fly ash were Fe oxides with very low Fe

solubilities.

**3.2.4 Biofuel burning aerosol**

For biofuel burning aerosol, Fe solubility in acetate buffer (pH: 4.3) ranges from 2.86%

to 100% with average and median values of 56.07 ± 30.95% and 55.87%, respectively (Table

2). Based on the relationship between combustion temperature and Fe-containing species in

emitted aerosols, Fe solubility was previously estimated at 35% for wood burning (i.e., biofuel)

aerosol (Rathod et al., 2020), in good agreement with our experimental results.

The biofuel examined in our experiment was burnt in a sealed stove and contained no

apparent local soil contamination. As such, these results are most representative of domestic

biofuel combustion for which the influence of soil-derived Fe can be expected to be negligible.

In contrast, wildfires represent dynamic open fire systems that emit aerosol Fe in both fine and

coarse fractions (Hamilton et al., 2019). During wildfire combustion, not only is the biofuel

(biomass) consumed, but local soils are also entrained into the smoke plumes (Hamilton et al.,

2022; Tegler et al., 2023). These soil-derived particles are typically larger (in particle size) and

less soluble than their biofuel-derived counterparts (Hamilton et al., 2022), resulting in a larger

mass of emitted Fe, albeit with a lower overall Fe solubility. Future studies would benefit from

capturing emissions from open burning scenarios to better characterize the properties of

wildfire-emitted Fe.





### 3.2.5 Municipal waste fly ash, oil fly ash and oil bottom ash


Fe solubility in acetate buffer (pH: 4.3) was determined to range from 0.58% to 2.41% for
municipal waste fly ash (Table 2), with average and median values being 1.51 ± 0.92% and
1.54%, respectively. Few previous studies measured Fe solubility for municipal waste fly ash.
Fe solubility was estimated to be <2% for municipal waste fly ash when combustion
temperature exceeded 1100 K (Rathod et al., 2020), agreeing with our experimental results.
Fe solubility in acetate buffer (pH: 4.3) was determined to be 11.70% and 13.43% for the
two oil fly ash samples we examined, with an average value of 12.56 ± 0.87%. In previous
work, Fe solubility was measured to be 35.7% at pH of 4.7 (Desboeufs et al., 2005) and 70%
in deionized water (Schroth et al., 2009), both higher than our results. Although Fe solubility
measured in different studies showed some variations, all the studies suggested that oil fly ash
exhibited very high Fe solubility. Moreover, Fe solubility in acetate buffer (pH: 4.3) was
measured in our work to be 25.47% for one heavy oil bottom ash.

### 3.2.6 Fe solubilities: comparison of anthropogenic and dust Fe


Figure 2b compares our measured Fe solubility for six types of combustion and
anthropogenic particles with that for desert dust. Biofuel burning aerosols (median: 55.87%),
residential coal combustion aerosols (median: 28.45%), and oil fly ash (median: 12.56%)
exhibited very high Fe solubility. Compared to desert dust, for which Fe solubility is around
0.5% (Chuang et al., 2005; Li et al., 2022b; Ooki et al., 2009; Schroth et al., 2009; Shi et al.,
2011), Fe solubility was also higher for municipal waste fly ash (median: 1.54%) but lower for
steelwork fly ash (median: 0.07%) and power plant coal fly ash (median: 0.03%).



Overall, Fe solubility in emitted particles was significantly higher for low-temperature
combustion (residential and biofuel burning aerosols) than high-temperature combustion
(municipal waste fly ash, steelwork fly ash, and power plant coal fly ash). This is because Fe
in emitted particles is mainly highly soluble Fe sulfates for low combustion (Bhargava et al.,
2009; Rathod et al., 2020) but Fe oxides with very low solubility for high temperature
combustion (Hu et al., 2006; Ram et al., 1995; Rathod et al., 2020). The outlier to this is oil fly
ash which was emitted by high temperature combustion but showed high Fe solubility. This is
probably because heavy oil has high sulfur content, leading to the formation of sulfuric acid in
combustion that can condense onto co-emitted Fe oxide particles and form highly soluble Fe
sulfate (Rathod et al., 2020; Sippula et al., 2009).
**3.3 Modeling Results**
Leveraging combustion Fe solubilities as measured in aerosol emitted from combustion
of different fuel sources (residential coal, industrial coal, oil, and residential biofuel), as
reported in Sections 3.1 and 3.2, we performed a series of Earth System Model simulations that
tested variable anthropogenic Fe solubilities at the point of emission. To pair observed
solubilities (Table 2) with fuel-types represented in the model, we updated Fe emission
solubility in industrial coal from 0.2 to 0.05%, in residential coal from 0.2 to 33%, in residential
biofuel burning from 10 to 56%, and in oil from 38 to 25% (Table 3). Smelting Fe solubility at
point of emission was kept at 0.03%, since new experimental data do not suggest an alternative
solubility. New solubility parameters were applied to both fine and coarse modes within MIMI.
A description of the fractional solubilities applied to each anthropogenic fuel type within each
model simulation is provided in Table 3.




**Table 3.** Fractional Fe solubilities applied in each model simulation to reflect experimental findings. Rows highlighted in gray indicate baseline simulations with no changes made to Fe solubility from previous work using MIMI. To underscore modifications between simulations, a dash (–) is provided where assigned solubility did not differ from the PD-BASE simulation.

| Simulation | Fe solubility modifications by fuel-type (%) | | | | |
| --- | --- | --- | --- | --- | --- |
| | Industrial Coal | Residential Coal | Oil | Residential Biofuel | Smelting |
| PD-BASE | 0.2 | 0.2 | 38 | 10 | 0.003 |
| PD-RESI | - | 33 | - | - | - |
| PD-BIOF | - | 33 | - | 56 | - |
| PD-IND | 0.05 | 33 | 25 | 56 | - |
| PI-BASE | NA | NA | NA | 10 | NA |
| PI-BIOF | NA | NA | NA | 56 | NA |
| MID-SSP370-BASE | 0.2 | 0.2 | 38 | 10 | 0.003 |
| MID-SSP370-BIOF | - | 33 | - | 56 | 0.003 |
| END-SSP370-BASE | 0.2 | 0.2 | 38 | 10 | 0.003 |
| END-SSP370-BIOF | - | 33 | - | 56 | 0.003 |

### 3.3.1 Impacts on global soluble Fe distribution

By updating Fe solubility for four distinct anthropogenic fuel-types in MIMI, our model simulations revealed a new range of potential soluble Fe fluxes to the modern ocean. When compared to PD-BASE, in the PD-RESI case soluble Fe fluxes to the surface ocean increased by 33 Gg a$^{-1}$ (92% increase) globally, with the largest increase in emissions stemming from China, India, Australia, the USA, central Europe, and South Africa (Figure 3). These increases follow previous reports of relatively large anthropogenic emissions from these regions when compared to global averages (Rathod et al., 2024; Wang et al., 2015).



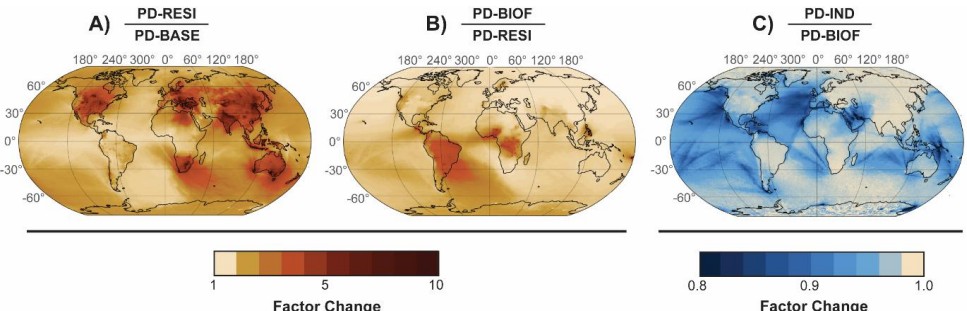

**Figure 3.** Relative changes to soluble Fe deposition fluxes following modifications to anthropogenic Fe solubilities in MIMI: A) residential coal, B) residential biofuel, and C) industrial (fossil fuels and oil). Note the difference in scales between A/B and C; red tones (A and B) indicate relative increases and blue tones (C) indicate relative decreases.

When we increased Fe solubility for residential biofuel in the model (PD-BIOF), soluble Fe fluxes to the ocean increased by an additional 11 Gg a$^{-1}$, totaling an increase of 44 Gg a$^{-1}$ compared to PD-BASE (Table 4). Changes to soluble Fe fluxes in this case were most concentrated across the South Atlantic (Figure 3), likely due to elevated emissions (as high as a factor of 4) along the eastern coast of South America and in sub-Saharan Africa where biofuel-burning in cook stoves is a common residential practice (García-López et al., 2025; Stoner et al., 2021). When both residential coal and biofuel Fe solubilities were increased (PD-BIOF), we report a maximum change in soluble Fe fluxes, with deposition to the ocean doubling from 40 to 80 Gg a$^{-1}$ at the global scale (123% increase relative to PD-BASE; Table 4). Both our minimum (PD-BASE) and maximum (PI-BIOF) estimates fell within previous ranges of uncertainty as reported for anthropogenic Fe deposition fluxes to the ocean (Hamilton et al., 2023; Ito and Miyakawa, 2023), suggesting that solubility modifications applied herein





aligns well with previous constraints for global soluble Fe fluxes by Earth System Models.
Altering Fe solubility for industrial coal (PD-IND) had a much smaller impact on soluble Fe
deposition fluxes to the ocean, only resulting in a 2.2 Gg a$^{-1}$ decrease (3% decrease) at the
global scale. These losses in fossil-fuel based Fe emissions were distributed across multiple
regions but most apparent within major shipping lanes (Figure 3).

**Table 4.** Global anthropogenic combustion Fe emission and deposition fluxes (Gg a$^{-1}$) in the
preindustrial (PI), present day (PD), and Future (FU), as simulated by MIMI.

| Emission Scenario | Model Simulation (case) | Fe Content | Global Emission | Global Deposition | Deposition to Ocean |
|---|---|---|---|---|---|
| PI (1750 CE) | PI-BASE | Total (Soluble) | 0.8 (0.08) | 0.8 (0.1) | 0.3 (0.04) |
| | PI-BIOF | Total (Soluble) | 0.8 (0.44) | 0.8 (0.5) | 0.3 (0.2) |
| PD (2010 CE) | PD-BASE | Total (Soluble) | 2220 (20) | 2220 (90) | 590 (40) |
| | PD-BIOF | Total (Soluble) | 2220 (170) | 2220 (270) | 590 (80) |
| FU (2050 CE) | MID-SSP370-BASE | Total (Soluble) | 2400 (20) | 2400 (90) | 620 (40) |
| | MID-SSP370-BIOF | Total (Soluble) | 2400 (180) | 2400 (250) | 620 (80) |
| FU (2100 CE) | END-SSP370-BASE | Total (Soluble) | 1970 (20) | 1970 (80) | 510 (30) |
| | END-SSP370-BIOF | Total (Soluble) | 1970 (90) | 1970 (150) | 510 (50) |



Modeled changes in residential coal emissions most impacted global soluble Fe fluxes
when compared to other fuel-types, and this was likely attributed to the fact that residential
coal Fe emissions (464 Gg a$^{-1}$) exceeded individual emissions from all other fuel types in our





PD emissions dataset (industrial coal: 305 Gg a$^{-1}$; oil: 34 Gg a$^{-1}$; residential biofuel: 72 Gg a$^{-1}$),
with the exception of smelting (1345 Gg a$^{-1}$). Furthermore, the solubility modification for
residential coal exceeded the relative changes to all other sectors (from 0.2% to 33%), resulting
in emission increase of soluble Fe by a factor of 10 in some regions (Figure 3).

**625    3.3.2 Model-observation comparisons of total and soluble Fe concentrations**

Comparison of modelled surface concentrations with regionally grouped, ship-based

observations revealed good agreement between modeled and observed total and soluble aerosol
Fe concentrations (Figure 4). Modeled total Fe concentrations fell slightly under observed
values but remained well within one order of magnitude for each ocean region, with the
exception of the Southern Ocean where total Fe was underestimated by several orders of
magnitude. Our underestimation of total aerosol Fe in the Southern Ocean aligns with previous
efforts to model global fluxes of total and soluble aerosol Fe using MIMI v1.0 and other Earth
System Models (Ito and Miyakawa, 2023; Ito et al., 2019; Liu et al., 2024). Current leading
hypotheses suggest that an Fe source, such as volcanism or mining, is not currently well
represented in models, or alternatively, that limited observations are not representative of
typical Fe conditions in the airshed of the Southern Ocean (Ito and Miyakawa, 2023; Liu et al.,

2024).



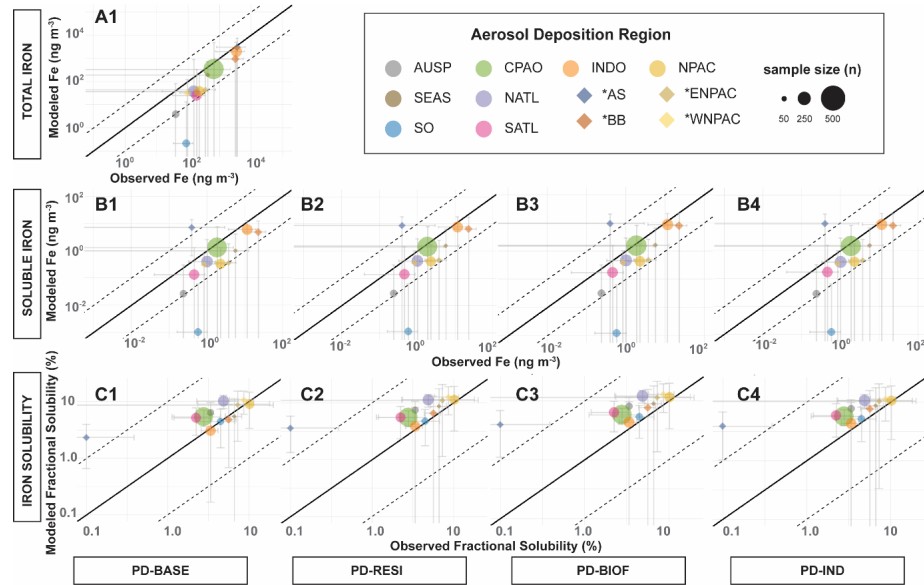

**Figure 4**. Comparison of modelling and observational data: A) total Fe, B) soluble Fe, and C)

Fe solubility; 1) PD-BASE, 2) PD-RESI, 3) PD-BIOF, and 4) PD-IND. Annual averages were

aggregated as medians and plotted by deposition region. Error bars represent spatiotemporal

variance within each region. The solid black line indicates a 1-to-1 relationship between

modeled and observed values and the dashed lines represent deviation by ±1 order of magnitude.

Only PD-BASE is shown for total Fe as total Fe fluxes were not manipulated between model

cases.

   In the PD-BASE case, for soluble Fe concentrations, regression analyses followed a

similar trend to total Fe, wherein for each deposition region, modeled averages were slightly

lower than observed values and fell within one order of magnitude, except for the Southern

Ocean and the Arabian Sea (Figure 4). In the PD-RESI and PD-BIOF cases, average soluble

Fe concentrations increased in every ocean region across the globe. At the global scale,



modifications made to anthropogenic Fe solubilities overall lessened the underestimation of
soluble Fe by MIMI (Figure 4). Increasing Fe solubility for residential coal burning emission
resulted in an average increase to modeled soluble Fe concentrations by $0.5 \pm 0.7$ ng m$^{-3}$ in
each ocean region (Table S8). Conversely, enhancing Fe solubility in residential biofuel
burning emission only improved model performance within the South Atlantic Ocean (Figure
4); solubility modifications to fossil fuel sources of Fe (industrial coal and oil emissions) did
little to improve model performance (Figure 4). Model performance improvements attributed
to soluble Fe in residential coal emissions were especially noticeable for Southeastern Asia, the
Bay of Bengal, the North Pacific, and the North Atlantic (Figure 4), likely because of the strong
source of residential coal combustion aerosol from China, India, and Europe (Figure 3).

Despite slight underestimations for total and soluble Fe by PD-BASE, Fe solubilities

calculated by the model were well aligned with observations for every region except for the
Arabian Sea, wherein solubility was overestimated by 1-2 orders of magnitude (Figure 4). In
previous MIMI-validation efforts (Hamilton et al., 2019), observational data from the Arabian
Sea and Bay of Bengal were aggregated as the Indian Ocean, but each basin has distinct
terrestrial sources of aerosol. While both receive a large source of anthropogenic emission from
India, the Arabian Sea has a much larger dust influence from the Middle East and the Bay of
Bengal is more strongly affected by anthropogenic emissions across Southeastern Asia (Bali et
al., 2019; Guieu et al., 2019).

In general, we found that ocean regions which are known to be influenced by dust Fe had

slightly higher overestimations for Fe solubility when compared to regions with less dust
deposition (Figure 4C). Specifically, Fe solubility in the South Atlantic (heavily influenced by





dust emitted from the Patagonia Desert), the Central Atlantic/Pacific and North Atlantic
(Sahara Desert), Australia/South Pacific (Great Victoria Desert), and North Pacific (Gobi
Desert), was overestimated when compared to other regions (Figure 4C). These findings
suggest that soluble Fe in dust is slightly overrepresented in the current model scheme (Figure
4). However, it is important to highlight that use of the new soil scheme that includes how soil
moisture changes impact dust emissions, as revealed herein, indicated an improvement from
previous model simulations for dust Fe (Hamilton et al., 2019; Myriokefalitakis et al., 2018).
In regions with less dust Fe and stronger sources of anthropogenic Fe, Fe solubilities were
slightly underestimated by the baseline simulation. In each of our modified solubility cases
(PD-RESI, PD-BIOF, and PD-IND), Fe solubility for southeastern Asia, the Bay of Bengal,
the North Pacific, and the Southern Ocean all increased, but the regions heavily impacted by
dust remained relatively unchanged (Figure 4). This suggests that modifications to Fe solubility
for anthropogenic combustion aerosol as applied herein improve the predictive capability of
MIMI in regions with heavy anthropogenic influence.
**3.3.3 Soluble Fe under PI and FU emission scenarios**
PI model simulations serve as a valuable baseline in understanding the specific
implications of anthropogenic perturbation on the Earth system. When comparing soluble Fe
emitted in the PI and PD, we found increases in soluble Fe fluxes for most regions across the
globe (Figure 5). Such increases were largely attributed to steadily growing anthropogenic
combustion and industrial activity emissions over time. However, dust and wildfire Fe
emissions were also distinct between the PI and PD, due to land-use changes and global
warming induced feedbacks that have altered global precipitation patterns (Hamilton et al.,




2018; Kok et al., 2023; Li et al., 2019; Mahowald et al., 2010). Globally, we estimated that
soluble Fe fluxes to marine ecosystems in the PD exceeded these during the PI era by 36-70
Gg a$^{-1}$, with the low estimate from PD-BASE and the high estimate from PD-BIOF case. We
observed the largest differences between PI and PD soluble Fe fluxes in Southeastern Asia, the
Bay of Bengal, and the North Atlantic. These differences were apparent for all the solubility
parameterizations used and deposition regions (Figures 5 and S4), apart from the South Atlantic
region where soluble Fe fluxes in the PI era exceeded PD fluxes by 5.6 Gg a$^{-1}$. This decrease
was likely attributed to reduction in wildfire burned area over past decades, particularly in
Africa (Andela et al., 2017; Jones et al., 2022). Previous work has suggested that wildfire
activity during the PI era exceeded current wildfire regimes at the global scale (Hamilton et al.,
2018), and our modeling work suggested larger wildfire Fe emission and deposition fluxes in
every region during the PI era compared to PD. This signal was also most apparent in the South
Atlantic where PD deposition fluxes of wildfire Fe were exceeded by a factor of 4 (Figure 5).

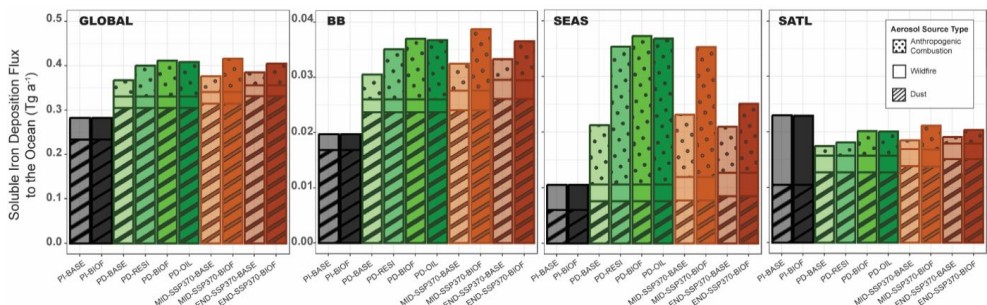


**Figure 5.** Deposition fluxes of soluble aerosol Fe to marine ecosystems: A) globally; B) Bay
of Bengal (BB); C) Southeastern Asia (SEAS); D) South Atlantic (SATL). Deposition fluxes
are source-apportioned (dust, wildfire burning, and anthropogenic combustion) and provided
for each case with distinct solubility parameters. Plots for each regional grouping (as depicted





in Figure 1) are provided in the supplementary information (Figure S4).

New Fe solubility parameterizations for residential coal emission increased the delivery of

anthropogenic soluble aerosol Fe to marine ecosystems by a factor of 1.7 – 2.8 at the regional
scale (Figures 3 and 5). Increased fluxes were greatest for Australia/the Southern Pacific, the
Southern Atlantic, across Southeastern Asia, the Bay of Bengal, and the Southern Ocean. While
model performance showed the most significant improvement in the Bay of Bengal,
enhancements in other regions were generally modest, and changes in the Southern Ocean were
indiscernible (Figure 4). These regional discrepancies highlight the current limitations of ship-
based observations in capturing representative soluble Fe fluxes, particularly in the under-
sampled Southern Hemisphere. Future efforts should prioritize expanding the spatial coverage
of measurements in these regions to improve model accuracy and understanding of possible
anthropogenic influence on remote marine biogeochemistry.

Under the SSP370 future emissions scenario, anthropogenic Fe emissions are projected to

reach their maxima by 2050 for most deposition regions and then decrease to values at or below
current PD conditions by 2100 (Figure 5). However, this trend did not hold for the global south
(Australia/South Pacific, Central Pacific/Atlantic, and Southern Ocean) where soluble aerosol
Fe fluxes were projected to continually increase through the end of the century (Figure 5); this
was not due to anthropogenic combustion emissions directly, but rather because of dust as the
primary source of soluble Fe to the Southern Hemisphere (Figure 5).

By the end of the century under SSP370, model simulations suggested that soluble Fe

deposition to marine systems will decrease by a factor of 0.2-0.4 globally, compared to the





present day. In our base simulation (PD-BASE), losses of soluble Fe are projected to be 7 Gg a$^{-1}$
by the year 2100. However, in each of our test cases, the projected changes to soluble Fe fluxes
between PD/MID-SSP370 and MID-SSP370/END-SSP370 was more drastic, with midcentury
soluble Fe fluxes increasing by 6 Gg a$^{-1}$ by 2050 but falling by up to 32 Gg a$^{-1}$ by the year 2100
(Figure 5). Therefore, marine ecosystems could face a more significant deviation from current
soluble Fe supplies than has been previously represented in Earth System Models, and this
could largely be attributed to changes in residential sources of combustion aerosol.

At the regional scale, the largest changes to soluble Fe fluxes under SSP370 are

anticipated in the Bay of Bengal, the North Atlantic, the South Atlantic, and across
Southeastern Asia (Figures 5 and S4). Notably, in Southeastern Asia, anthropogenic
combustion exceeded both dust and wildfire Fe, constituting up to 72% of all soluble Fe fluxes
to marine ecosystems in this region. By 2100, soluble Fe fluxes were projected to decrease by
a factor of 0.2 in the base case (PD-BASE vs. END-SSP370-BASE; Figure 5). However, in the
-BIOF cases for these scenarios, fluxes decreased by a factor of 0.5, more than doubling the
projected reduction in soluble Fe fluxes. This would constitute a much larger disruption to
current biogeochemistry and external sources of Fe in key open ocean regions.

Projected losses in soluble Fe fluxes by 2100 under future emission scenarios, including

SSP370, have strong implications for the spatiotemporal distribution of net marine primary
productivity, especially in Fe limited regions. Recent work suggested that the atmospheric
supply of anthropogenic Fe has already shifted phytoplankton bloom dynamics in the open
ocean by accelerating the seasonal uptake of upwelled nitrogen in HNLC regions, such as the
North Pacific (Hawco et al., 2025). Such HNLC regions are anticipated to be especially



sensitive to changes in anthropogenic Fe given that they are historically limited by trace metals
including Fe (Bazzani et al., 2023; Moore et al., 2013; Nishioka and Obata, 2017).

Diverse lines of evidence suggest that half of the soluble Fe flux to the North Pacific

comes from Asian anthropogenic sources (Hamilton et al., 2019; Hamilton et al., 2020b;
Hawco et al., 2025; Rathod et al., 2020). Li et al. (2024) found that the magnitude of
chlorophyll-a response to Fe deposition off the coast of China was lowered by a factor of 4
during COVID-19 in March 2020 when anthropogenic emissions across East Asia were
substantially reduced. The authors speculated that a reduction in soluble Fe from anthropogenic
activities, either via the primary emission of soluble Fe or via a reduction in Fe solubilization
via co-emitted acidic species (e.g., $SO_x$), resulted in a lessened supply of soluble Fe delivered
during the deposition event. Moreover, using Fe isotopes to trace source origins of atmospheric
Fe, Hawco et al. (2025) recently showed that the springtime delivery of anthropogenic Fe could
be one major factor driving observed seasonal and geographic shifts to the North Pacific
transition zone, a highly productive boundary in the North Pacific. In particular, our findings
suggest that residential coal burning is an especially important source of soluble Fe to the North
Pacific and the South China Sea, and across southeastern Asia. Accordingly, we find that
projected losses of anthropogenic emissions over the course of this century will greatly
influence nutrient dynamics in these key marine ecosystems.
**4 Conclusions**

Anthropogenic activity has added a multitude of new aerosol Fe sources to the atmosphere.

Understanding how these new sources alter soluble Fe fluxes aids better understanding of
changes to marine primary productivity and ocean ecosystems within the Anthropocene.



However, estimating the contribution of anthropogenic emissions to soluble aerosol Fe fluxes
is challenging given the wide variety of sources, leading to large variation across different
modeling studies on the magnitude of the deposition fluxes. By measuring Fe content and
solubility of aerosol Fe from several important anthropogenic sources and including a first
assessment of the contribution from biofuels, we find that median Fe solubilities vary by greater
than three order of magnitude, from 0.03% for power plant coal fly ash to 55.87% for biofuel
burning aerosol.

To understand the impact of increasing source representation of fractional Fe solubilities

measured in this work, we refined Fe solubility parameters within an atmospheric Fe module
(MIMI) embedded within the CESM2. We found that current (PD) and projected (FU) fluxes
of soluble aerosol Fe to global marine ecosystems could exceed current modeled values by up
to a factor of 2, mainly driven by the new addition of a highly soluble biofuel Fe source,
highlighting residential burning as a significant source of soluble Fe to the ocean. The most
notable impacts were found in the Bay of Bengal, across Southeast Asia, and throughout the
North Pacific and North Atlantic (i.e., regions strongly influenced by nearby continental
anthropogenic activity). However, many shipborne observations of aerosol Fe have historically
been taken in regions outside of biofuel rich Fe plumes, limiting current capacity to constrain
model fluxes and highlighting a future research opportunity.

**Data availability.**

Experimental data can be found in the manuscript or the supplement. Modeling output

data, coding scripts, and emission inventories are available at:



https://github.com/haleyplaas/CombustionFe.

**Competing interests.**

At least one of the (co-)authors is a member of the editorial board of Atmospheric Chemistry and Physics.

**Author contribution.**

MT initiated this study; MT and DSH designed this study and secured funding resources; RL, YZ, YC and TZ conducted experimental work; HEP, SR and DSH conducted modeling work; YY provided key samples used in this work and contributed to data analysis; RL and HEP analyzed the results; RL, HEP, DSH and MT wrote the manuscript; all the authors reviewed and approved the manuscript.

**Financial support.**

This work was sponsored by National Natural Science Foundation of China (42321003, 42277088 and 22361162668), International Partnership Program of Chinese Academy of Sciences (164GJHZ2024011FN), Guangzhou Bureau of Science and Technology (2024A04J6533), and Guangdong Foundation for the Program of Science and Technology Research (2023B1212060049).



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
