# Peer review of "Residential burning is a significant source of soluble iron to the ocean"

_EGUsphere, 2025_

## Author Comment (AC1)

Comments by referees are in blue.
Our replies are in black.
Changes to the manuscript are highlighted in red both here and in the revised manuscript.

**Reply to referee #1**

There are still large uncertainties in source attribution of soluble iron (Fe) in atmospheric aerosols from different anthropogenic sources, despite substantial advancement in the source attribution between natural and anthropogenic sources. Due to low iron content in residential burning aerosols, residential burning is expected to be negligible source of Fe to the ocean. This study argues that residential burning is a significant source of soluble Fe to the ocean based on the measurements of high Fe solubility at emission for residential coal burning aerosol and residential biofuel burning than industrial coal fly ash. This hypothesis is based on the assumption on a representation of iron mineralogy for anthropogenic combustion sources, which suggests that the emissions of soluble iron from residential coal and wood combustion could be an important contributor to soluble iron production (Rathod et al. 2020). The comprehensive measurements and their application to the model simulations may help us to advance modeling anthropogenic soluble iron. However, my major concern is the inconsistency between Fe content used in the model and Fe content measured from this work with the high Fe solubility at emission measured in this work. I have some comments and questions to improve this paper.

**Reply:** We thank ref #1, Dr. Ito, for his thoughtful comments and helpful suggestions, which have reshaped how we address the question of the role of residential iron in the supply of soluble iron to the ocean from the atmosphere. Following the his suggestion, we have conducted new analyses that quantifies additional uncertainties in soluble Fe fluxes due to the representation of residential iron in the emissions inventory. The uncertainty revealed that residental updates increase anthropogenic fluxes to the ocean by between 25% and 100% depending on Fe content assumptions. Please refer to our revised manuscript and supplement as well as our reply to specific comments (as shown below) for further details.

We would like to point out that our conclusion that residential burning is a significant source of soluble Fe is not based on the work by Rathod et al. (2020); instead, this conclusion is based on our experimental measurements of Fe solubility for several important anthropogenic and combustion sources.

Major comments:

1. With the combination of the high Fe solubility measured in this work and Fe content used in the model, higher Fe solubility might be applied to residential burning particles at higher Fe content with different Fe species in the revised model. Since the model results presented in this paper may represent an upper bound estimate of soluble Fe at emission, it is highly recommended to conduct additional simulations using both the Fe solubility and Fe content measured in this work.

**Reply:** We agree that our first set of simulations represent an upper-bound estimate of soluble Fe fluxes from anthropogenic combustion sources. To understand how Fe content in the emissions inventory also impacted results, we have undertaken a new set of simulations that quantified the impact of residential sources using low Fe content data. We have revised the language throughout the reported results to reflect this new analyses.

Overall, model–observation comparisons were relatively unchanged by the choice of Fe content in residental Fe sources (Figure 4 in the revised manuscript, page 36). Furthermore,

modeled soluble Fe concentrations remain underestimated compared to obervations in key anthropogenically impacted regions of the atmosphere. This suggests that either data set is suitable for use, but more importantly highlights a need for further observations into the varied roles of the multiple anthropogenic Fe sources.

2. Industrial-to-residential BC is higher than industrial-to-residential Fe, because of higher carbon content and lower iron content for residential sector than industrial sector. It is straightforward to calculate the Fe-to-BC ratios for residential sector and industrial sector from Table S6 in Rathod et al. 2020. If the same industrial-to-residential BC ratios were applied to industrial-to-residential Fe, this must be corrected. My recommendation is to calculate industrial and residential Fe emissions separately and use them in the model.

**Reply:** We agree additional simulations were necessary to bound this uncertainty in examining Fe content for residential coal Fe emissions. Following this recommendation, we have generated a new inventory (which we label as the "low-residential emissions") and we performed three additional simulations using this inventory to quantify the impact of different assumptions in Fe content. The ratios were calculated using sector-specific Fe emissions from Table S6 in Rathod et al. (2020) and sector-specific BC emissions from Hoesly et al. (2018) that were scaled using the more detailed sector-specific BC sources in Bond et al. (2004). This method was chosen to maintain spatial heterogeneity of anthropogenic sources as now outlined in the methods. Ultimately, we found that when applying both emissions inventories, soluble iron fluxes differed by 25% to 100% at the global scale, aligning with previous reports for anthropogenic Fe uncertainty (Hamilton et al., 2023).

While Dr. Ito is kindly referred to our revised manuscript and supplement for more details, we summarized major changes are below:

1) The data used to make the new emissions inventory is now described in Table S8 (page 17 in the revised supplement), and the final emissions by sector are reported in Table S9 (page 18 in the revised supplement).

2) Development and implementation of the new emissions inventory, is now detailed in section 2.2.2 (page 11-14 in the revised manuscript).

Specific comments

1. L.116: Please describe the information about the size of ash samples.

**Reply:** As suggested, we measured size distribution of our fly samples using dynamic particle scattering. The volume-mean diameters are mostly tens of μm for fly ash samples, while residential coal and biofuel combustion aerosols are PM2.5 samples (as mentioned in the first paragraph of Section 2.1.2). In the revised manuscript (page 6), we have added one senetence to provide size information for fly ash samples we used: "The volume-mean diameters, determined using diameter light scattering, were found to be 16.9-67.6, 4.7-176.4, 21.2-115.9 and 15.4 μm for power plant coal fly ash (n=31), steelwork fly ash (n=29), municipal waste fly ash (n=3), and oil fly ash (n=1) samples, respectively (Li, 2025)."

2. L.121 and Table S5: Please specify the sources of oil fly ashes, because Fe content is substantially different between the two samples. Why is this?

**Reply:** In the revised manuscript (page 7) we have made the following change to include further information for the two oil fly ash samples: "…two oil fly ash samples which were PM$_{2.5}$ samples emitted by heavy oil and diesel fuel combustion in the engine of a cargo ship (Wu et al., 2018)…" Moreover, in the revised SI (page 9), we have updated Table S5 to provide further details for the two oil fly ash samples and the three municipal waste fly ash samples we examined.

The Fe content differed by a factor of ~2 for the two oil fly ash samples. On the other hand, such a difference is not surprisingly large, as the Fe content differed by a factor of ~5 for the 31 power plant coal fly ash samples we examined. The number of oil fly ash samples we examined is very small, precluding us from further explanation.

The numbers of samples examined in our work are very small for municipal waste fly ash and oil fly/bottom ash, and the results may be not be statistically representative. Therefore, we have moved relevant results and discussion to the revised supplement (page 3-4), and updated the manuscript accordingly.

3. L.129: You described that the acetate buffer was used to simulate cloud water (Li et al., 2022b). Here, you applied Fe solubility from the acetate buffer to that at emission before the cloud processing. Why did you choose the sodium acetate buffer over the deionized water? Please show the comparison of Fe solubility near the source regions, for example, at Guangzhou.

**Reply:** Indeed a wide range of leaching solutions and leaching protocols have been used in previous studies. We chose sodium acetate buffer (instead of ultrapure water), mainly because its pH did not change during leaching due to its larger buffering capacity than ultrapure water. In the revised manuscript (page 8) we have added the following sentence to provide further explanation: "A wide range of protocols, differing in leaching solution, filter pore size, and so on, were used to extract soluble Fe, and the results obtained using different leaching protocols could be substantial (Tang et al., 2025). Sodium acetate buffer, instead of ultrapure water, was used in the present work as the leaching solution, because its pH did not change during leaching due to its larger buffering capacity compared to ultrapure water."

Fe solubility was found to vary over a very large range (from 0.03% for power plant coal fly ash to ~56% for biofuel burning aerosol). Given such a large variation, the comparison with Fe solubility near source regions may not be very meanful.

4. L.198: You updated Fe solubility at emission but did not Fe content. I suggest additional simulations with both updated Fe content and solubility at emission (see below more details).

**Reply:** Following this recommendation, we have generated a new inventory (which we label as the "low-residential emissions") and we performed three additional simulations using this inventory. The ratios were calculated using sector-specific Fe emissions from Table S6 in Rathod et al. (2020) and sector-specific BC emissions from Hoesly et al. (2018) that was scaled using sector-specific BC sources in Bond et al. (2004). This was chosen to maintain spatial heterogeneity of anthropogenic sources as now outlined in the methods

5. L.214: The anthropogenic coal Fe sources from residential sector can be separately estimated from industrial sector, as is described on L.383. Do you mean that you segregated 1) industrial fossil fuel (coal) in the dataset provided by Rathod et al. (2020) into industrial and residential coal burning sources? Please rephrase the sentence.

**Reply:** In the revised manuscript (Section 2.2.2, page 11-14) we detailed development and implementation of the new emissions inventory. We kindly refer Dr. Ito ro our response to his major comment #2 for more details.

6. L.216: How did you assume that the Fe-to-BC ratios were matched between sources? Iron emissions from residential wood and coal combustion is only about 2% (Table S6 in Rathod et al. 2020) of the global fine Fe emissions due to their low-Fe emission factors. Thus, the industrial-to-residential BC is higher than industrial-to-residential Fe (see Table 3 in Ito et al. 2021), as you described for carbon content on l.386 and iron content in Figure 2. If you used the same Fe-to-BC ratios between the two sources, you assigned larger Fe emissions (in other

words, higher Fe content than Rathod et al., 2020) into residential sector than industrial sector by more than a factor of 10. It is straightforward to calculate the Fe-to-BC ratios for residential sector and industrial sector because the anthropogenic coal Fe sources from residential sector is separately estimated from industrial sector (Rathod et al., 2020). Please show the global emissions for Fe and BC from coal combustion for each sector and final Fe content at emission.

**Reply:** We thank Dr. Iro for this comment, and agree that by assuming residential-to-industrial Fe tracks residential-to-industrial BC, as reported in the CMIP6 dataset, we may overestimate residential coal-Fe in our emissions inventory (now reported as "high-residential"). Following his recommendation, we have now updated the original residential coal-Fe inventory to create a second "low-residential" emission dataset that follows the Fe content by fuel source values reported in Rathod et al. (2020). Residential coal Fe emissions for each inventory are reported in Table S9 (page 18 in the revised supplement).

However, even using the inventory that may have overstated residential Fe, results for global soluble Fe fluxes only varied by a factor of 2 (global soluble Fe deposition was ~130 Gg a$^{-1}$ using the "low" inventory, whereas fluxes were ~270 Gg a$^{-1}$). This uncertainty is now reported in Table 5 (page 35 in the revised manuscript).

7. L.320: Please indicate the references for the additional measurements.

**Reply:** In the revised manuscript (page 19) we have added the following sentence to indicate the references for the additional measurements: "The observational dataset of Fe content in aerosol was reported in Hamilton et al. (2019) and updated herein to include measurements from Srinivas et al. (2012) and more recent studies published between 2021 and 2024 (n = 1624) (Desboeufs et al., 2024; Elliott et al., 2024; Kurisu et al., 2021; López-García et al., 2021; Marafante et al., 2024; Panda et al., 2022; Perron et al., 2022; Rodríguez et al., 2021; Sakata et al., 2022; Seo and Kim, 2023; Winton et al., 2022; Wu et al., 2023; Zhang et al., 2024)."

8. L.383: Table S2 shows negative relationship between Fe content and Fe solubility, possibly because Fe species inside the particles might not be transformed to labile form in combustion process. If you used higher Fe content (> 0.5 mg/g) for aerosols, you would apply larger Fe emissions (> 13 times) to residential coal combustion aerosols which consist of the median Fe content (0.038 mg/g) measured at higher Fe solubility. Why don't you update Fe content, too?

**Reply:** Specific comment #8 is addressed together with specific comment #10, as detailed below.

9. L.347, Table 2: Please indicate the size information and elucidate the differences between ash and aerosol.

**Reply:** As suggested, we measured size distribution of our fly samples using dynamic particle scattering. The volume-mean diameters are mostly tens of μm for fly ash samples, while residential coal and biofuel combustion aerosols are PM2.5 samples (as mentioned in the first paragraph of Section 2.1.2). In the revised manuscript (page 6), we have added one senetence to provide size information for fly ash samples we used: "The volume-mean diameters, determined using diameter light scattering, were found to be 16.9-67.6, 4.7-176.4, 21.2-115.9 and 15.4 μm for power plant coal fly ash (n=31), steelwork fly ash (n=29), municipal waste fly ash (n=3), and oil fly ash (n=1) samples, respectively (Li, 2025)."

10. L.430: If you used higher Fe content (0.58 mg/g) for aerosols, you would apply larger Fe emissions (45 times) to residential biofuel combustion aerosols which consist of the median Fe content (0.013 mg/g) measured at higher Fe solubility. Why don't you update Fe content, too?

**Reply:** Here we address specific comments # 8 and #10 together. We thank Dr. Ito once more for raising this important point. We have now updated Fe content as suggested.

Application of the new emissions inventory results in soluble Fe fluxes to the global ocean, when compared to our high-residential emissions inventory (49 vs 81 Gg a⁻¹), representing an approximate global factor of 2 uncertainty in the magnitude of fluxes due to uncertanties in residental fuel charecteristics. Details of new flux ranges are included in the revised supplement (Tables S10a and S10b, page 19-20).

11. L.568 and Table 3: Why did you use higher Fe solubility of oil bottom ash (25%) than the measurements of Fe solubility in Table 2 for oil fly ash (12.56%)?

**Reply:** We chose 25% due to a lack of statistically significant data in our study measurements (n=2 and n=1). Given that the measured lower (12.56%) value is much lower than previously reported solubilities of oil we chose 25% as a test to understand what impacts reducing the fractional solubity might have. Give the comment, we have re-evaluated the choice to explore oil in this paper that is focused on residential Fe, and removed all reference of oil from the main paper so as not to cause further confusion for readers.

12. L.658: Please report the statistics for soluble Fe over Southeastern Asia, the Bay of Bengal, the North Pacific, and the North Atlantic to support the improvement.

**Reply:** We agree that reporting the summary statistics are necessary to support our claims with Figure 4.

In response to this comment, we have provided a new table (Table S12 in the revised supplement, page 22-23) that includes median and average surface concentrations, variances, and root mean square errors (RSME) for each case, grouped by region

Morever, we have reported the reductions in RMSE between cases for key regions in the revised manuscript (page 37): "For regions most influenced by residential coal burning, the improvement in model skill was slightly higher using the high-emissions inventory, especially for Southeastern Asia (ΔRMSE -0.5), the Bay of Bengal (ΔRMSE = -5.0), and the eastern North Pacific (Figure 4; Table S12; ΔRMSE = -0.1). For biofuel burning Fe, the emissions inventory had no effect, but enhancing Fe solubility most improved model skill within the South Atlantic Ocean (Figure 4, Table S12 ΔRMSE = -0.07). Complete summary statistics conveying impacts to model skill for soluble Fe concentrations simulated in each run are provided in Table S12 of the Supplemental Information."

13. L.678: Please report the statistics for Fe to support the improvement of dust Fe.

**Reply:** RMSE comparisons between model versions with and without the new soil state scheme are now reported in Table S9 (page 18 in the revised manuscript).We have chosen to provide this information in the supplement, as the focus of this work is on anthropogenic combustion Fe.

In addition, in the revised manuscript (page 10) we have made the following change to describe how we improve dust Fe: "Following the implementation of a new soil-moisture scheme, dust was rescaled to attain a global climatological mean dust aerosol optical depth of ~0.03 (Ridley et al., 2016), consistent with all previous versions of the MIMI model. The inclusion of these improvements to dust and updated anthropogenic Fe sources represents a new working version of MIMI v1.1, as described herein, and detailed validation efforts are reported in the Supplement (Figure S3 and Table S11)."

14. L.685: Please report the statistics for Fe solubility.

**Reply:** Following the same additions for summary stats for soluble Fe, this is now reported in Table S13 of the revised supplement (page 24-25).

15. l.768: Since you increased soluble Fe from anthropogenic sources, you would estimate lighter Fe isotopes than your previous estimates. How can this be reconciled with the aerosol Fe isotope measurements? If higher Fe solubility is compensated by lower Fe content in the

**Reply:** The Fe isotopic endmembers for residental coal vs industrial coal and for residential wood burning vs. open biomass burning are both currently unknown, but would be needed for such an analysis. The Fe isotope data that has been published and used to contrain the MIMI model to date has also been focused on the Central North Atlantic (Conway et al. 2019) and North Pacific (Bunell et al. 2025); both regions where the residental Fe contribution to soluble iron is relatively low (Figure 3). We therefore agree that understanding anthropogenic Fe isotopes, from a wide variety of sources, as a way to help fingerprint anthropogenic Fe and contrain the model is an interesting avenue of future research, but far outside the scope of this paper.

In the revised manuscript (page 43) we have made the following change to further discuss this issue: "Moreover, using Fe isotopes to trace source origins of atmospheric Fe, Hawco et al. (2025) recently showed that the springtime delivery of anthropogenic Fe could be one major factor driving observed seasonal and geographic shifts to the North Pacific transition zone, a highly productive boundary in the North Pacific. Isotopic signatures capable of distinguishing residential coal combustion from other anthropogenic combustion sources have not yet been identified, but our findings suggest that residential coal burning is an especially important source of soluble Fe to the North Pacific and the South China Sea, and across southeastern Asia. Accordingly, we find that projected losses of anthropogenic emissions over the course of this century will most greatly influence nutrient dynamics in these key marine ecosystems."

**Reply:** We meant concentration. In the revised manuscript (page 4) we have changed it to "The quantity of Fe in ocean waters plays a particularly important role…"

**Reply:** Thank you for catching this labeling error. We have actually removed this from the main text, and this label is explained in the supplement.

---

## Author Comment (AC2)

Comments by referees are in blue.
Our replies are in black.
Changes to the manuscript are highlighted in red both here and in the revised manuscript.

**Reply to referee #2**

This manuscript presents new measurements of iron (Fe) content and solubility across a range of anthropogenic fuel sources, with a particular focus on residential combustion (coal and biofuel). These observations are subsequently integrated into the MIMI–CESM2 model framework to assess impacts on soluble Fe deposition to the ocean. The study addresses a critical knowledge gap in quantifying anthropogenic aerosol Fe emissions and their influence on ocean biogeochemistry, especially in high-nutrient, low-chlorophyll (HNLC) regions. The experimental dataset is extensive and covers key combustion sources in China, while the modeling analysis provides valuable constraints on the climatological and future patterns of soluble Fe deposition. The finding that residential combustion disproportionately contributes to soluble Fe fluxes to the ocean is novel and has significant implications for anthropogenic Fe cycling and marine productivity. The manuscript is generally well-written, well-structured, and scientifically solid. However, several points require clarification and enhanced discussion before the manuscript can be recommended for publication.

**Reply:** We would like to thank ref #2 for reviewing our manuscript and recommending it for publication after minor revision. We have carefully addressed all the comments, as detailed below.

Major comments:

1. Line 136–137: Please provide the detection limit for soluble Fe. In addition, briefly describe the QA/QC procedures for the measurement method to ensure data reliability.

**Reply:** The detection limit of Fe was 0.5 μg/L. In the revised manuscript (page 9) we have added one section (Section 2.1.3) to provide information for the detection limit and the major QA/QC procedures we used: "The detection limit of Fe in solutions was determined to be 0.5 μg/L in this work. A reference solution (NIST 1643f) was used to check the accuracy of ICP-MS analysis, and the difference between actual and measured concentrations was found to be <1%. Furthermore, three blanks (with no fly ash or filters not loaded with any particles) were used in each batch when we measured total or soluble Fe. The background levels of soluble Fe were always below the detection limit; the background levels of total Fe, ranging from 4.3-5.7 μg/L, were much lower than total Fe concentrations for most of our samples and subtracted when we reported our results."

2. Line 448–490: While the manuscript highlights the importance of acid processing in promoting Fe dissolution, the evidence supporting atmospheric processing-induced release of soluble Fe from power-plant coal fly ash is not clearly demonstrated. Please provide more explicit evidence or discussion.

**Reply:** We think here ref #2 refers to Line 488-490, instead of Line 488-490. As we stated in the original manuscript (Line 484-490), previous laboratory studies (Chen et al., 2012; Fu et al., 2012; Baldo et al., 2022) found that acid processing could significantly increase promote Fe dissolution of coal fly ash. We may not understand very well what ref #2 means; in this case, we are happy to make further revision if ref #2 can explain this comment more specifically.

3. Line 556–559: The statement "This is because Fe in emitted particles is mainly highly soluble Fe sulfates for low combustion" requires clarification. What is meant by "low

**Reply:** In our original manuscript we made a typo, and "..low combustion.." should have been "low temperature combustion". In the revised manuscript (page 30) we have corrected it.

In our original manuscript (Section 3.2.2, page 498-502) we have provided a few sentences to explain why combustion at low and high temperatures will lead to aerosol Fe with different solubility: "Pyrite ($FeS_2$) is the major Fe-containing mineral in coal (Deng et al., 2015; Oliveira et al., 2016; Rathod et al., 2020). In low-temperature combustion, pyrite is mainly transformed to Fe sulfate (Bhargava et al., 2009) which has very high Fe solubility; as the temperature increases to >1000 K, Fe sulfate is further transformed to hematite and magnetite which exhibit very low solubility (Hu et al., 2006; Ram et al., 1995; Rathod et al., 2020)." As a result, we feel that it is not necessary to repeat this explanation in Section 3.2.6.

4. Line 590–594: Figure 3b shows that the largest relative increases in soluble Fe deposition occur primarily over equatorial/tropical regions such as the Congo Basin and Amazon rainforest. Does this imply that enhanced soluble Fe deposition over the South Atlantic originates largely from long-range transport of soluble Fe associated with South American residential biofuel combustion? Please clarify this interpretation.

**Reply:** Yes, this is our interpretation of Figure 3 as well. In the revised manuscript (page XX) we have made the following change to enhance our discussion: "Changes to soluble Fe fluxes from biofuel burning were most concentrated across the South Atlantic (Figure 3), likely due to the long range transport of emissions from the Amazon rainforest and across the Congo River basin where biofuel-burning in cook stoves is a common residential practice (García-López et al., 2025; Stoner et al., 2021)."

5. Line 638–645: Figure 4 indicates a systematic model underestimation of total Fe (A1) and a systematic overestimation of soluble Fe (B1–B4), leading to an overestimate of aerosol Fe solubility. Please explain the potential causes of these biases and implications for the modeled Fe dissolution scheme.

**Reply:** There are several considerations when examining Figure 4 that need to be taken into account. First, the model-observation comparison uses observations that represent a single snapshot of the atmospheric state from highly transient and spatially sparse shipborne campaigns. Often this can mean we have a single day observation representing 1000's of kilometers. As a full spatiotemporal data set is not available, the goal here is not to attain a perfect model-observational match but to gain a level of constraint at the regional level that is acceptable given the large uncertainty. The only region that the model – observation range for total (and thus soluble) does not represent well is the Southern Ocean. However, Southern Ocean solubility is not well captured – an issue other Fe models struggle with (see Myriokefalitakis et al. 2018).

Second, the figure itself is a function of the chosen region definition over which to calculate medians. Here we have chosen "classic" physical ocean definitions. However, if the regions were modified to instead represent aerosol source provinces the results will look different (e.g., the recent evaluation by Bergas-Masso et al. (2025)). This is once again due to the sparse data being used.

Third, the ultimate goal of the MIMI model is to capture soluble iron, as that is the parameter used for biogeochemical modelling. In Figure 4 it can be seen that for all regions, apart from the Southern Ocean, the model-observation relaionship fall slightly low, but are relatively aligned with each other. This suggests that the spatial relationship between regions is robust, but the magnitude itself may be low. This, coupled with the low bias in total iron,

suggests that there may be a missing source of Fe in the model rather than any dissolution chemistry bias.

One conclusion of this paper is that residental iron does not fill that knowledge gap. But moreover, we have a dataset of observations that do not well capture downwind residental iron plumes and suggest that this is considered in future campaigns.

6. The comparison with ship-based observations is useful; however, as noted by the authors, some ship measurements are outside biofuel-influenced regions. Such data should not be included for model validation in this specific context, as they do not reflect the emission regime of interest.

**Reply:** In lieu of source-apportioned observations, we replotted the model-observation comparison data only including observations collected in ocean regions downwind of strong residential iron influences, which we defined as areas where soluble Fe fluxes increased by a factor of 2 or more in the PD-BIOF simulations. However, this resulted in dropping our n value from n = 990 to n = 25 (media aggregated data within 25 grid cell locations), so this approach was ultimately not desirable for model evaluation purposes.

Those plots using the "biofuel-influenced' regions are now provided in Figure S4 of the revised supplement (page 15); in addition, in the revised manuscript (page 38) we have added one paragraph to discuss this further: "Withstanding source-apportioned measurements of residential coal or biofuel aerosol in our observational dataset, we performed additional model-observation comparisons only using measurements collected in ocean regions downwind of strong residential burning influences. These regions were defined as model-resolution grid cells wherein soluble Fe fluxes increased by 100% or more in the PD-BIOF simulations (Figure 3). However, this reduced median-aggregated observational data points from n=990 to n=25, limiting statistical capacity to constrain model fluxes. When using the smaller observational dataset, model-observational comparisons for Total Fe, soluble Fe, and solubility mirrored agreement trends using the larger dataset (Figure 4); those regression analyses are provided in the Supplemental Information with results from the PD-IND simulation (Figures S4-S5)."

Minor comments

1. l.116: Please report the effective number of valid filter samples for each biofuel type in the main text.

**Reply:** We feel that such information may not be critical, and therefore we include it in Tables S2 and S4. In the revised manuscript (page 8) we have made the following change to specifically refer reader to the supplement: "…the number of effective filter samples (for which Fe content and solubility were reported) were usually <8 for each fuel type (see Tables S2 and S4 for further information)."

2. Line 74–75: grammatical issue with "to and from?" — please revise.

**Reply:** In the revised manuscript (page 4) we have change "…to and from the atmosphere and surface ocean..." to "…from the atmosphere and to the surface ocean…" to make it clearer.

3. Line 97: clarify whether "aerosol solubility" refers specifically to Fe solubility.

**Reply:** Yes, it refers to Fe solubility. In the revised manuscript (page 5) we have changed "…Fe content and solubility of aerosol emitted by…" to    "…Fe content and solubility for aerosol emitted by…", in order to be clearer.

4. Line 114: revise "in order…..." to "in order to……".

**Reply:** In the revised manuscript (page 9) we have changed "…in order…" to "…in order to …".

5. Several minor grammatical issues exist; a careful proofreading is recommended.

**Reply:** We would like to thank ref #2 for the careful review. As suggested, during our revision we have carefully checked the entire manuscript in order to minimize typos and defer any remaining oversights to the editorial team.

---

## Author Comment (AC3)

Comments by referees are in blue.
Our replies are in black.
Changes to the manuscript are highlighted in red both here and in the revised manuscript.

**Reply to referee #3**

This manuscript presents an important and timely topic regarding the role of anthropogenic combustion sources, particularly residential burning, in supplying soluble iron (Fe) to the ocean. The study combines direct laboratory measurements of Fe solubility across a wide range of anthropogenic fuels with the MIMI model. It provides a thorough investigation into the relatively contributions from anthropogenic activities for the present-day and pre-industrial era. The manuscript is generally well written, well referenced, and of interest to the atmospheric chemistry, ocean biogeochemistry, and Earth system modeling communities. Therefore, I would recommend it to be accepted with minor revisions, if the following comments can be properly addressed.

**Reply:** We would like to thank ref #3 for reviewing our manuscript and recommending it for publication after minor revision. We have carefully addressed all the comments, as detailed below.

Major comments:

1. A major innovative contribution of this study is quantifying the contributions of Fe from multiple main sources with newly measured emission factors and modeling simulations. The authors mention ranges of solubility and provide multiple simulations. While there are several processes that may retain large uncertainties with the modeling effort (e.g., Fe-to-BC ratio variability, emission inventory biases, solubility parameter assumptions), the propagation of uncertainties is not fully quantified. A sensitivity analysis is recommended to improve the reliability of the conclusions. For example, how much do uncertainties in residential coal solubility affect global estimates? A sensitivity analysis may help provide more robust policy-relevant implications for the conclusions.

**Reply:** This is an important point that was also raised by ref #1. Accordingly, we have reworded much of methods section 2.2.2 to better describe inventory development and have additionally created/tested an additional Fe emissions inventory that follows different assumptions to separate residential from industrial sources of Fe aerosol. Using the new inventory (now labeled as a high-residential [the original] and low-residential emissions inventory), we conducted 3-additional model simulations in the present day to compare and assess the resultant uncertainties stemming from residential burning.

We report these findings in a sensitivity analysis as suggested and interpretation has been updated in Section 3.3. Ultimately, the uncertainty revealed that residental updates increase anthropogenic fluxes to the ocean by between 25% and 100% depending on Fe content assumptions. Ref #3 is kindly referred to our revised manuscript and supplement for more details on our revision, and below we summarize major changes we have made:

1) The data used to make the new emissions inventory is now described in Table S8 (page 17 in the revised supplement), and the final emissions by sector are reported in Table S9 (page 18 in the revised supplement).

2) Development and implementation of the new emissions inventory, is now detailed in section 2.2.2 (page 11-14 in the revised manuscript).

3) The uncertainty resulting from the emissions inventory and its assumptions are now shown in Figure 3 (page 34 in the revised manuscript), Figure 4 (page 36-37 in the revised manuscript) and Figure 5 (page 40 in the revised manuscript).

Minor comments

1. In the abstract it would be better to briefly describe the relative importance of residential burning compared to other anthropogenic sources (e.g., percentage contribution to total soluble Fe flux).

**Reply:** As suggested, we have revised the abstract accordingly (page 3 in the revised manuscript): "Anthropogenic combustion is estimated to contribute up to 20% of the global soluble Fe flux to the ocean in the present day. Furthermore, we identified residential coal burning as a previously neglected but potentially important source with regional flux contributions ranging from <1% to 21%. Our work underscores the need to further refine understanding of aerosol Fe properties from a wide variety of anthropogenic sources by increasing observations in more novel aerosol regimes, with a focus on residential coal burning. This understanding will in turn aid in characterizing the influences of anthropogenic activity on past, present, and future atmospheric nutrient inputs to marine ecosystems."

2. line#83: "A study … is needed" or "studies … are needed".

**Reply:** The referee is right. In the revised manuscript (page 5) we have revised this sentence to "…studies examining socioeconomic, technology, and policy driven changes to anthropogenic fuel-burning, are needed to…"

3. line#473 "was found to range" should be "ranged".

**Reply:** As suggested, in the revised manuscript (page 27) we have changed it to "Fe solubility in acetate buffer (pH: 4.3) ranged from 0.002% to 0.17%..."